

# Fairness-enhancing classification methods for non-binary sensitive features—*How to fairly detect leakages in water distribution systems*

Janine Strotherm, Inaam Ashraf and Barbara Hammer

Center for Cognitive Interaction Technology, Universität Bielefeld, Bielefeld, North Rhine-Westphalia, Germany

## ABSTRACT

Especially if artificial intelligence (AI)-supported decisions affect the society, the fairness of such AI-based methodologies constitutes an important area of research. In this contribution, we investigate the applications of AI to the socioeconomically relevant infrastructure of water distribution systems (WDSs). We propose an appropriate definition of protected groups in WDSs and generalized definitions of group fairness, applicable even to multiple non-binary sensitive features, that provably coincide with existing definitions for a single binary sensitive feature. We demonstrate that typical methods for the detection of leakages in WDSs are unfair in this sense. Further, we thus propose a general fairness-enhancing framework as an extension of the specific leakage detection pipeline, but also for an arbitrary learning scheme, to increase the fairness of the AI-based algorithm. Finally, we evaluate and compare several specific instantiations of this framework on a toy and on a realistic WDS to show their utility.

# INTRODUCTION

Due to the increasing usage of artificial intelligence (AI)-based decision making systems in socially relevant fields of application, the question of *fair decision making* gained much importance in recent years (cf. *Angwin et al., 2016*; *European Union, 2019*). Fairness is hereby related to the several (protected) groups or individuals, which are affected by the algorithmic decision making and characterized by *sensitive features* such as gender or ethnicity. Most algorithms on which these tools are based rely on data which can be biased with respect to questions of fairness without intention, resulting in skewed models. Also, the algorithm itself can discriminate against protected groups or individuals without explicitly aiming to do so due to an undesirable algorithmic bias (cf. *Mehrabi et al., 2021*; *Pessach & Shmueli, 2022*). This gives rise to the question of how to define fairness and how to mitigate unfairness in case it occurs in the context of machine learning (ML), *i.e.*, in the context of data-driven algorithms.

Corresponding author
Janine Strotherm,
jstrotherm@techfak.uni-bielefeld.de

**Background: Fairness definitions** Several definitions of fairness as well as approaches to achieve these fairness standards have been theoretically discussed and tested in practice (cf. *Barocas, Hardt & Narayanan, 2019*; *Castelnovo et al., 2022*; *Dwork et al., 2012*; *Mehrabi et al., 2021*; *Pessach & Shmueli, 2022*). From a legal perspective, one distinguishes between *disparate treatment* and *disparate impact* (DI) (cf. *Barocas, Hardt & Narayanan, 2019*). While disparate treatment occurs whenever a group or an individual is intentionally treated differently because of their membership in a protected group, disparate impact is a consequence of indirect discrimination happening despite "seemingly neutral policy" (cf. *Pessach & Shmueli, 2022*).

From a scientific viewpoint, the variety of fairness notions is much larger where many popular approaches focus mainly on (binary) classification tasks (cf. *Castelnovo et al., 2022*; *Mehrabi et al., 2021*; *Pessach & Shmueli, 2022*). Different definitions can be grouped into the concepts of *group fairness*, *individual fairness*, *causal fairness* and *dynamic fairness*: Group fairness aims at treating different groups equally while individual fairness aims at treating similar individuals similarly. Causal fairness examines the extent to which the sensitive feature, such as gender or ethnicity, influences the prediction of a model and dynamic fairness examines the long-term effects of (supposedly) fair decisions (cf. *Strotherm et al., 2023*).

The fairness notions that we will discuss in this work belong to the former concept of group fairness. Here, most works focus on fairness definitions with respect to a single binary sensitive feature that splits the underlying population into a *discriminated* and a *privileged* group (cf. *Feldman et al., 2015*; *Hardt, Price & Srebro, 2016*; *Kamiran & Calders, 2009*, *2010*; *Mehrabi et al., 2021*; *Pessach & Shmueli, 2022*; *Ruf & Detyniecki, 2021*; *Zafar et al., 2017a*, *2017b*). There is some work on fairness definitions based on the independence assumption of the model's prediction and a single non-binary sensitive feature; however, there is no rigorous theory on how this assumption translates to generalized fairness notions as necessary and sufficient conditions of this independence assumption and their relation to the binary case (cf. *Agarwal et al., 2018*; *Castelnovo et al., 2022*). We will build on this point.

**Background: Fairness methods** Besides the definition of fairness, the problem arises as to how to enhance fairness in well-known ML methods while maintaining a reasonable overall performance of the model. Approaches can hereby be grouped into three categories: Depending on when in the training pipeline the model is enhanced with respect to fairness, we speak about pre-, in- or post-processing techniques (cf. *Barocas, Hardt & Narayanan, 2019*; *Mehrabi et al., 2021*; *Pessach & Shmueli, 2022*).

Pre-processing methods usually modify the training data which is fed to the training algorithm. For example, *Kamiran & Calders (2010)* use a resampling technique by removing unpreferred samples, *i.e.*, positive outcomes in the privileged group and negative outcomes in the discriminated group, and duplicating preferred samples, *i.e.*, positive outcomes in the discriminated group and negative outcomes in the privileged group, that lie close to the decision boundary of a binary classifier. In another work, they modify the training data by changing the labels of training samples that lie close to the decision boundary of a binary classifier such that negative outcomes in the

privileged group and positive outcomes in the discriminated group appear more often (cf. *Kamiran & Calders, 2009*). While these methods aim at putting more emphasis on the discriminated group and less emphasis on the privileged group, *Feldman et al. (2015)* modify the non-sensitive features of training samples such that it is not able to predict the sensitive feature from the non-sensitives. This reduces the chance that the model's predictions, which are based on the non-sensitive features, are correlated with the sensitive feature.

In contrast, post-processing methods modify the model after the training. For example, *Pleiss et al. (2017)* modify a pre-trained model by randomly changing some outputs of a binary classifier on the group on which the classifier performs better to ensure equal performance over all groups. As another example, *Hardt, Price & Srebro (2016)* retrain a pre-trained model by optimizing a loss between the new and the pre-trained binary classifier while satisfying fairness-constraints. Another simple approach is to use group-specific thresholds for a threshold-based classifier (cf. *Corbett-Davies et al., 2017*).

Finally, in-process methods modify the (original) training algorithm. A common way to do so is by adding fairness-constraints (cf. *Agarwal et al., 2018*; *Agarwal, Dudík & Wu, 2019*; *Calders et al., 2013*; *Komiyama et al., 2018*; *Narasimhan et al., 2020*; *Zafar et al., 2017a*, *2017b*) or a fairness-regularization-term (cf. *Aghaei, Azizi & Vayanos, 2019*; *Berk et al., 2017*; *Pessach & Shmueli, 2022*) to the loss function that is to be optimized. Next to classification, also regression tasks usually fall into this category (cf. *Agarwal, Dudík & Wu, 2019*; *Aghaei, Azizi & Vayanos, 2019*; *Berk et al., 2017*; *Calders et al., 2013*; *Komiyama et al., 2018*; *Narasimhan et al., 2020*). The methods presented in this work are also in-process methods and are extensions of our work, *Strotherm & Hammer (2023)*, published in Springer's Lecture Notes in Computer Science. Both of these works are based on the methods of *Zafar et al. (2017b)*, but adapted to more generalized settings, as we will elaborate in the contributions paragraph.

**Background: Fairness in water distribution systems (WDSs)** The question of fairness becomes especially relevant when the decisions of an ML model impact socioeconomic infrastructure, such as WDSs. To the best of our knowledge, our previous work, *Strotherm & Hammer (2023)*, has been the first approach to introduce fairness within this domain. In that work, *Strotherm & Hammer (2023)*, we address the important problem of leakage detection in WDSs and investigate how far typical models treat different groups of consumers of the WDS (in)equally, and we will extend these considerations in this work as outlined in the next paragraph. As an extended version, portions of this work were previously published as part of the previous version (cf. *Strotherm & Hammer, 2023*).

**Contributions** Our approaches to improve group fairness in such a domain of high social and ethical relevance are based on the idea of considering the locality in the WDS as a sensitive feature. Considering the empirical covariance between the sensitive feature(s) and the model's prediction as a proxy for the fairness measure, similar to *Zafar et al. (2017b)*, but also the generalized fairness notions directly, are the base of all our proposed methods. The advantage of our fairness-enhancing algorithms is that they can handle even multiple non-binary sensitive features and satisfy both the concept of disparate treatment *and* disparate impact simultaneously, which is an asset towards most fairness-enhancing

algorithms (cf. *Pessach & Shmueli, 2022*; *Zafar et al., 2017b*). In more detail, our contributions—also in view of what this extension offers compared to our previous work, *Strotherm & Hammer (2023)*—are as follows:

- We propose group fairness definitions even for multiple non-binary sensitive features, which are generalizations of well-known corresponding fairness notions in the common setting of a binary classifier and a single binary sensitive feature.
- As an extension to our previous work, we provide details on the mathematical concept of independence, derive easy-to-test independence criteria, and leverage these in order to derive those generalized group fairness definitions. Moreover, we prove that those coincide with the aforementioned well-known corresponding fairness notions.
- We introduce a common leakage detection pipeline and propose a suitable definition of sensitive features and group fairness in the context of leakage detection in WDSs, with more detail in this work compared to our previous work. Consecutively, we present specific and already existing instantiations of this pipeline and show that common leakage detection methods do not obey these fairness criteria, with one more specific instantiation (based on the more powerful graph convolutional network (GCN) based virtual sensors instead of linear regression based virtual sensors) and with more detail in this work compared to our previous work.
- We introduce a fairness-enhancing leakage detection framework as an extension of the common leakage detection pipeline, with more detail in this work compared to our previous work. Consecutively, we present specific instantiations of this framework, among others by modifying the ideas of *Zafar et al. (2017b)* to any (ensemble) classification model instead of a convex margin-based binary classifier, to propose several fairness-enhancing methods, with more specific instantiations, among others based on the ideas in our previous work made on possible modifications of our methodologies.
- We provide an empirical evaluation of our proposed methods. As an extension to our previous work, next to the application of these methods to the toy WDS *Hanoi*, we investigate the application to the more complex and realistic WDS *L-Town*.

**Structure of the work** The rest of this work is structured as follows: In section "Group fairness in machine learning", we introduce definitions of group fairness for multiple non-binary sensitive features, giving the mathematical background for the derivation of such generalized definitions and how they are connected to already existing definitions. Afterwards, in section "Leakage detection in water distribution systems", we present a standard methodology to detect leakages in WDSs, introduce the meaning of sensitive features in this context and investigate whether the resulting model makes fair decisions with respect to the previously defined notions of fairness. Consecutively, in section "Fairness-enhancing leakage detection in water distribution networks", we propose and evaluate several adaptations of this methodology that enhance fairness and provide empirical evidence for our theoretical findings regarding the equivalence of different

fairness notions in this specific domain of application. Finally, our findings are summarized and discussed in section "Conclusion".

# GROUP FAIRNESS IN MACHINE LEARNING

On an abstract level, the concept of group fairness is based on the mathematical concept of (conditional) independence of two random variables (cf. *Barocas, Hardt & Narayanan, 2019*; *Castelnovo et al., 2022*). Therefore, in this section, we will first investigate this concept of independence in general (subsection "Independence of two random variables"). Consecutively, we will introduce the mathematical notation required to define an ML task and its group fairness based on this general concept of independence (subsection "Mathematical notation for machine learning") to be able to derive group fairness definitions in generalized ML tasks, which coincide with well-known definitions in more specific settings (subsection "Generalized notions of group fairness in machine learning").

## Independence of two random variables

As it is the main mathematical concept to characterize different notions of group fairness, for the sake of convenience, we recapitulate the concept of independence of two random variables in this subsection. Moreover, we target an easy, necessary and sufficient condition for this concept, which is particularly simple to apply and test in the context of fairness of ML models. Hence, we derive an equivalent formulation, lemma 2.2, which can be tested on canonical subsets of the full $\sigma$-fields.

For the rest of this subsection, let $(\Omega, \mathcal{F}, \mathbb{P})$ be a probability space, $(\mathcal{X}, \mathcal{F}_{\mathcal{X}})$, $(\mathcal{Y}, \mathcal{F}_{\mathcal{Y}})$ measurable spaces and $X : \Omega \to \mathcal{X}$, $Y : \Omega \to \mathcal{Y}$ random variables[1].

**Definition 2.1** (Independence of two random variables (cf. *Bauer, 1996*)). *X* and *Y* are independent with respect to the probability measure $\mathbb{P}$, if the $\sigma$-fields[2]
$\sigma(X) := \{X^{-1}(A) \mid A \in \mathcal{F}_{\mathcal{X}}\} \subset \mathcal{F}$ and $\sigma(Y) := \{Y^{-1}(B) \mid B \in \mathcal{F}_{\mathcal{Y}}\} \subset \mathcal{F}$ generated by these random variables are independent with respect to $\mathbb{P}$.

Based on that, in Appendix A.2, we derive general necessary and sufficient conditions for independence of two random variables. In the context of fairness of ML models, we are usually interested in a more specific setting, namely the independence of two discrete random variables.

**Lemma 2.2** (Independence of two discrete random variables). *Assume that X and Y are discrete, i.e., that $\mathcal{X} = \{x_1, \ldots, x_{K_x}\}$ and $\mathcal{Y} = \{y_1, \ldots, y_{K_y}\}$ holds. Then X and Y are independent with respect to $\mathbb{P}$ if and only if (iff)*

$$\mathbb{P}(X = x \mid Y = y_{k_1}) = \mathbb{P}(X = x \mid Y = y_{k_2}) \tag{2.1}$$

*holds for all $x \in \mathcal{X}$ and $y_{k_1}, y_{k_2} \in \mathcal{Y}$ for which $\mathbb{P}(Y = y_{k_1}), \mathbb{P}(Y = y_{k_2}) > 0$ holds.*

*Proof.* $\mathcal{E}_{\mathcal{X}} = \{\emptyset, \{x_1\}, \ldots, \{x_{K_x}\}\}$ and $\mathcal{E}_{\mathcal{Y}} = \{\emptyset, \{y_1\}, \ldots, \{y_{K_y}\}\}$ are $\cap$-stable generators of the $\sigma$-fields $\mathcal{F}_{\mathcal{X}} = P(\mathcal{X}) = \sigma(\mathcal{E}_{\mathcal{X}})$ and $\mathcal{F}_{\mathcal{Y}} = P(\mathcal{Y}) = \sigma(\mathcal{E}_{\mathcal{Y}})$, respectively. Therefore, by remark A.9 and lemma A.12 (we can replace the $\sigma$-fields $\mathcal{F}_{\mathcal{X}}$ and $\mathcal{F}_{\mathcal{Y}}$ in lemma A.12 by their generators $\mathcal{E}_{\mathcal{X}}$ and $\mathcal{E}_{\mathcal{Y}}$, respectively), *X* and *Y* are independent iff

$$\mathbb{P}(X \in \{x\} \mid Y \in \{y_{k_1}\}) = \mathbb{P}(X \in \{x\} \mid Y \in \{y_{k_2}\})$$

[1] The statements in this subsection do not only hold for random variables (*i.e.*, $\mathcal{X} \subset \mathbb{R}$, $\mathcal{Y} \subset \mathbb{R}$), but also for random vectors (*i.e.*, $\mathcal{X} \subset \mathbb{R}^{d_x}$, $\mathcal{Y} \subset \mathbb{R}^{d_y}$) and even for random elements (*i.e.*, $\mathcal{X}$ and $\mathcal{Y}$ are arbitrary sets). We just use the term random *variable* as this is the more familiar term.

[2] Of course, it requires a proof to show that these set systems are $\sigma$-fields. We give more information on this topic in Appendix A.1 and especially, lemma A.4.

holds for all $x \in \mathcal{X} = \{x_1, ..., x_{K_x}\}$ and all $y_{k_1}, y_{k_2} \in \mathcal{Y} = \{y_1, ..., y_{K_y}\}$ for which $\mathbb{P}(Y \in \{y_{k_1}\}), \mathbb{P}(Y \in \{y_{k_2}\}) > 0$ holds. Note that we can omit the cases $X \in \emptyset$ and $Y \in \emptyset$, as these are trivially fulfilled.

Lemma 2.2 guarantees independence of two discrete random variables by only testing one-elementary events. As the $\sigma$-fields $\mathcal{F}_{\mathcal{X}} = P(\mathcal{X})$ and $\mathcal{F}_{\mathcal{Y}} = P(\mathcal{Y})$, given by all subsets of $\mathcal{X}$ and $\mathcal{Y}$, respectively, consist of many more non-trivial events, lemma 2.2 gives us a valuable necessary and sufficient condition for independence of two discrete random variables, which we will make use of in the setting of ML.

## Mathematical notation for machine learning

Our next goal is the mathematical definition of group fairness as a formalization of equal treatment of an ML model *independent of sensitive attributes*. Such attributes, also called *sensitive features*, provide information about the membership or non-membership of a protected group, such as gender or ethnicity, to which the model should not exhibit any prejudice (cf. *Mehrabi et al., 2021*; *Pessach & Shmueli, 2022*). A later goal of this work will be to find a reasonable meaning of sensitive information in the context of WDS.

To formalize this independence and derive easy-to-test notions of group fairness based on the previous subsection "Independence of two random variables" in the next subsection "Generalized notions of group fairness in machine learning", we need to introduce mathematical notation that allows us to consider independence of two random variables in the context of ML. In such context, probabilities such as in subsection "Independence of two random variables" appear for random variables such as the model's output $\hat{Y} : \Omega \to \mathcal{Y}$, the labels $Y : \Omega \to \mathcal{Y}$, the features $X : \Omega \to \mathcal{X}$ or, in case of *fair* ML, the sensitive features $S : \Omega \to \mathcal{S}$ with target space $\mathcal{Y}$, feature space $\mathcal{X}$ and sensitive feature space $\mathcal{S}$.

In this work, we will consider a one-dimensional binary classification task and a single discrete but possibly non-binary sensitive feature[3], *i.e.*, the target space equals $\mathcal{Y} = \{0, 1\} \subset \mathbb{N}_0$ and the (finite) sensitive feature space equals $\mathcal{S} = \{s_1, ..., s_K\} \subset \mathbb{N}_0$. Equipping each with the power set makes them a measurable space.

**Example 2.3** (Getting an intuition on fair ML). The domain $\Omega$ could consist of criminals in the US and the model $\hat{Y}$ could predict whether $(\hat{Y}(\omega) = 1)$ or whether not $(\hat{Y}(\omega) = 0)$ a criminal $\omega \in \Omega$ will be criminal again in the future. This prediction should be independent of their ethnicity $S(\omega) \in \mathbb{N}_0$ (cf. *Angwin et al., 2016*).

The typical goal of ML is to learn the relation between the features $X$ and the labels $Y$, *i.e.*, either the distribution $\mathbb{P}(X, Y)^{-1} = \mathbb{P}((X, Y) \in \cdot)$ of $(X, Y)$ (generative ML) or more often, the distribution $\mathbb{P}(Y \in \cdot \mid X = x)$ of $Y$, given $X$, (discriminative ML) for any $x \in \mathcal{X}$.[4] However, as these distributions are usually unknown, we use training data, *i.e.*, samples[5]

$$\mathcal{D} = \{(x_i, y_i) \in \mathcal{X} \times \mathcal{Y} \mid i = 1, ..., n\} = \{(X, Y)(\omega_i) \in \mathcal{X} \times \mathcal{Y} \mid i = 1, ..., n\}$$

to estimate $\mathbb{P}X^{-1}$ by $\widehat{\mathbb{P}}X^{-1} := \frac{1}{n}\sum_{i=1}^{n}\delta_{x_i}$, *etc.* When it comes to fairness, we extend the training data by the sensitive attribute:

$$\mathcal{D} = \{(x_i, s_i, y_i) \in \mathcal{X} \times \mathcal{S} \times \mathcal{Y} \mid i = 1, ..., n\}.$$

[3] Generalizations to multi-dimensional non-binary classification or regression tasks and multiple non-binary or continuous sensitive features are possible. While we will not go into detail regarding continuous labels or sensitive features, we will provide information on generalizability for the other cases.

[4] An arbitrary random variable $X : \Omega \to \mathcal{X}$ as given in subsection "Independence of two random variables" induces a measure $\mathbb{P}X^{-1}$ on $\mathcal{F}_{\mathcal{X}}$ by $\mathbb{P}X^{-1}(A) := \mathbb{P}(X \in A)$ for all $A \in \mathcal{F}_{\mathcal{X}}$.

[5] Samples or statistical units can be both the elements $\omega \in \Omega$ from the population space as well as the realisations $(x, y) = (X, Y)(\omega) \in \mathcal{X} \times \mathcal{Y}$ of the random vector $(X, Y)$.

Next to these distributions, often, a functional relation between the features $X$ and the labels $Y$ is the object of interest. This is done by learning the overall model $\hat{Y}$, composed of a learnable model (or model function) $f : \mathcal{X} \to \mathcal{Y}$, applied to the features $X : \Omega \to \mathcal{X}$. In such a case, we consider a hypothesis space $\mathcal{H} = \{f : \mathcal{X} \to \mathcal{Y} \mid f \in \mathcal{H}\}$, *i.e.*, a (sub-) space of functions mapping from the feature space $\mathcal{X}$ to the target space $\mathcal{Y}$. Consecutively, we want to learn the relation between $X$ and $Y$ by finding the optimal function $f \in \mathcal{H}$, such that $\hat{Y} := f(X) \approx Y$ holds. In most cases, the hypothesis space is a set of functions $\mathcal{H} = \{f_\Theta : \mathcal{X} \to \mathcal{Y}, x \mapsto f_\Theta(x) \mid \Theta \in \mathbb{R}^{d_p}\}$ parameterized by a parameter $\Theta \in \mathbb{R}^{d_p}$.

Finally, learning the functional relation between the features $X$ and the labels $Y$ by learning the optimal model $\hat{Y} \approx Y$ is done by comparing the results $\hat{y}_i = f_\Theta(x_i)$ of the model $\hat{Y} = f_\Theta(X)$ to the desirable results $y_i$ for all $i = 1, \dots, n$ from the training data $\mathcal{D}$. The comparisons are done by using a suitable loss function which is applied to these magnitudes and optimized with respect to the parameter(s) that characterize(s) $f_\Theta$.

*Remark 2.4.* Note that often, in ML-related literature, the introduction of $\Omega$ is omitted. Instead, random variables $X$ on $\mathcal{X}$, $Y$ on $\mathcal{Y}$, *etc.*, are introduced. We introduce $\Omega$ to guarantee a well-defined usage of probabilities such as $\mathbb{P}(X = x)$ for some $x \in \mathcal{X}$, *etc.*

## Generalized notions of group fairness in machine learning
### *Motivation*

Reflecting that there is no unique definition of fairness in real life, there is an enormous amount of different definitions of fairness in ML. While focusing on group fairness, even this category can be further grouped into three subcategories: Independence[6], separation and sufficiency. In this context, group fairness can be characterized by some independence connected to the (binary) classification model $\hat{Y} : \Omega \to \mathcal{Y}$, the true label $Y : \Omega \to \mathcal{Y}$, and the sensitive feature $S : \Omega \to \mathcal{S}$. *Barocas, Hardt & Narayanan (2019)* define these concepts as follows: *Independence* requires (mathematical) independence between the model's prediction $\hat{Y}$ and the sensitive feature $S$. *Separation* requires independence between the model's prediction $\hat{Y}$ and the sensitive feature $S$, conditioned on events based on the label $Y$. *Sufficiency* requires independence between the label $Y$ and the sensitive feature $S$, conditioned on events based on the model's prediction $\hat{Y}$. In this work, we will focus on the usually harder to achieve concepts of independence and separation.

However, there are also other definitions that fall under the broad umbrella of group fairness, but which can also be sorted in one of these subcategories (cf. *Ruf & Detyniecki, 2021*). They are usually defined for a one-dimensional binary classification task and for a single binary sensitive feature only, *i.e.*, in settings where $\mathcal{Y} = \mathcal{S} = \{0, 1\}$ holds (cf. *Mehrabi et al., 2021*; *Pessach & Shmueli, 2022*; *Ruf & Detyniecki, 2021*). For example, one well-known fairness definition is called *disparate impact*. In most literature, it is assumed that $\{Y = 1\}$ is the class of interest and $\{S = 0\}$ is the discriminated group, and therefore, $\mathbb{P}(\hat{Y} = 1 \mid S = 0) < \mathbb{P}(\hat{Y} = 1 \mid S = 1)$ holds. In this case, the disparate impact score is defined as the proportion of the *passing rate* of the discriminated group from the privileged group

[6] Mind the difference between the mathematical concept of independence and the fairness concept of independence. It is usually clear from the context what of both concepts is meant.

$$\text{DI} := \frac{\mathbb{P}(\hat{Y} = 1 \mid S = 0)}{\mathbb{P}(\hat{Y} = 1 \mid S = 1)} \tag{2.2}$$

and should satisfy $\text{DI} \geq 1 - \epsilon$ or $\text{DI} \geq \frac{p}{100}$ for some $\epsilon \in [0, 1]$ or $p \in [0, 100]$ (cf. *Pessach & Shmueli, 2022*). The latter rule is also known as the $p\%$-rule, and $p = 80$ (or $\epsilon = 0.2$) is a desirable choice (cf. *Pessach & Shmueli, 2022*; *Zafar et al., 2017b*). At the same time, the 80%-rule is also a popular legal term and the reason that the disparate impact score received its importance: It is "designed to mathematically represent the legal notion of *disparate impact*" (cf. *Pessach & Shmueli, 2022*), which requires to avoid that "one group's passing rate is less than 80% of the group with the highest rate" (cf. *Biddle, 2006*).

The goal of the rest of this subsection is on the one hand to connect these different group fairness notions and on the other hand to introduce generalized notions for more general settings. More precisely, our contribution to this existing research is as follows:

- Starting from the definitions of *Barocas, Hardt & Narayanan (2019)* for independence (subsection "Independence") and separation (subsection "Separation") each, we will make use of the particularly easy necessary and sufficient condition for the independence of two random variables (lemma 2.2), which we derived in subsection "Independence of two random variables" to derive easy-to-test and generalized notions of group fairness in the context of subsection "Mathematical notation for machine learning". In detail, these notions are applicable for more general settings, *i.e.*, not only for a one-dimensional binary classification task and a single binary sensitive feature, which is the setting on which the majority of the literature focuses (cf. *Feldman et al., 2015*; *Hardt, Price & Srebro, 2016*; *Kamiran & Calders, 2009*; *Kamiran & Calders, 2010*; *Mehrabi et al., 2021*; *Pessach & Shmueli, 2022*; *Ruf & Detyniecki, 2021*; *Zafar et al., 2017a, 2017b*).
- Based on these notions, we will derive generalized empirical notions of the most common group fairness definitions…
- … and prove that these coincide with the corresponding definitions in the setting of a one-dimensional binary classification task and a single binary sensitive feature.

While the generalized empirical group fairness definitions will appear to be intrinsic compared to already existing definitions, our theoretical work in subsection "Independence of two random variables" shows that these generalizations display not only a necessary but a sufficient condition for the desired independence criterion on which they are based. As a summary, an overview of already existing definitions and how we extend these is displayed in Tables 1 and 2 (subsection "Summary of generalized notions of group fairness").

For technical reasons, we assume that all of the following conditional probabilities exist.

### *Independence*
**An easy-to-test notion of group fairness**
**Definition 2.5** (Fairness according to the independence criterion (cf. *Barocas, Hardt & Narayanan, 2019*)).

**Table 1 Overview of exact fairness definitions.** Overview of our exact derived necessary, sufficient and easy-to-test fairness conditions (corollary 2.6 and 2.13) based on the corresponding definitions of *Barocas, Hardt & Narayanan (2019)* (definition 2.5 and 2.12).

| | Definition according to Barocas et al. | Derived necessary and sufficient condition |
|---|---|---|
| Independence | $\hat{Y} \perp\!\!\!\perp S$ | $\mathbb{P}(\hat{Y} = y \mid S = s_{k_1})$ $=$ $\mathbb{P}(\hat{Y} = y \mid S = s_{k_2})$ $\forall y \in \mathcal{Y}, s_{k_1}, s_{k_2} \in \mathcal{S}$ |
| Separation | $\hat{Y} \perp\!\!\!\perp S \mid Y$ | $\mathbb{P}(\hat{Y} = \hat{y} \mid S = s_{k_1}, Y = y)$ $=$ $\mathbb{P}(\hat{Y} = \hat{y} \mid S = s_{k_2}, Y = y)$ $\forall y, \hat{y} \in \mathcal{Y}, s_{k_1}, s_{k_2} \in \mathcal{S}$ |

**Table 2 Overview of empirical fairness definitions.** Comparison of our generalized empirical fairness definitions (definition 2.7, 2.8, 2.14 and 2.15) and the corresponding existing definitions (*e.g.*, *Pessach & Shmueli, 2022*).

| | Derived generalized empirical definitions (multi cases) | Existing empirical definitions (binary cases) |
|---|---|---|
| Independence: DI | $\mathcal{Y} = \{0,1\}, \mathcal{S}$ arbitrary: $\min_{s_{k_1}, s_{k_2} \in \mathcal{S}} \frac{\mathbb{P}(\hat{Y}=1 \mid S=s_{k_1})}{\mathbb{P}(\hat{Y}=1 \mid S=s_{k_2})}$ | $\mathcal{Y} = \mathcal{S} = \{0,1\}$: $\frac{\mathbb{P}(\hat{Y}=1 \mid S=0)}{\mathbb{P}(\hat{Y}=1 \mid S=1)}$ |
| Independence: DP | $\mathcal{Y}, \mathcal{S}$ arbitrary: $\max_{y \in \mathcal{Y}, s_{k_1}, s_{k_2} \in \mathcal{S}}$ $\left\| \mathbb{P}(\hat{Y} = y \mid S = s_{k_1}) - \mathbb{P}(\hat{Y} = y \mid S = s_{k_2}) \right\|$ | $\mathcal{Y} = \mathcal{S} = \{0,1\}$: $\left\| \mathbb{P}(\hat{Y} = 1 \mid S = 0) - \mathbb{P}(\hat{Y} = 1 \mid S = 1) \right\|$ |
| Separation: EO | $\mathcal{Y} = \{0,1\}, \mathcal{S}$ arbitrary: $\max_{s_{k_1}, s_{k_2} \in \mathcal{S}}$ $\left\| \mathbb{P}(\hat{Y} = 1 \mid S = s_{k_1}, Y = 1) - \mathbb{P}(\hat{Y} = 1 \mid S = s_{k_2}, Y = 1) \right\|$ | $\mathcal{Y} = \mathcal{S} = \{0,1\}$: $\left\| \mathbb{P}(\hat{Y} = 1 \mid S = 0, Y = 1) - \mathbb{P}(\hat{Y} = 1 \mid S = 1, Y = 1) \right\|$ |
| Separation: EOs | $\mathcal{Y}, \mathcal{S}$ arbitrary, $\forall y \in \mathcal{Y}$: $\max_{s_{k_1}, s_{k_2} \in \mathcal{S}}$ $\left\| \mathbb{P}(\hat{Y} = y \mid S = s_{k_1}, Y = y) - \mathbb{P}(\hat{Y} = y \mid S = s_{k_2}, Y = y) \right\|$ | $\mathcal{Y} = \mathcal{S} = \{0,1\}$: $\left\| \mathbb{P}(\hat{Y} = 1 \mid S = 0, Y = 1) - \mathbb{P}(\hat{Y} = 1 \mid S = 1, Y = 1) \right\|$ $\left\| \mathbb{P}(\hat{Y} = 1 \mid S = 0, Y = 0) - \mathbb{P}(\hat{Y} = 1 \mid S = 1, Y = 0) \right\|$ |

The classification model $\hat{Y}$ is fair with respect to the sensitive feature $S$ in the sense of the independence criterion if and only if (iff) $\hat{Y}$ and $S$ are mathematically independent with respect to $\mathbb{P}$.

Based on this definition of *Barocas, Hardt & Narayanan (2019)*, lemma 2.2 induces the following easy-to-test independence criterion in the context of fair ML:

**Corollary 2.6** (Fairness according to the independence criterion).

*The classification model $\hat{Y}$ is fair with respect to the sensitive feature S in the sense of the independence criterion iff*

$$\mathbb{P}(\hat{Y} = y \mid S = s_{k_1}) = \mathbb{P}(\hat{Y} = y \mid S = s_{k_2})$$

*holds for all $y \in \mathcal{Y} = \{0,1\} \subset \mathbb{N}_0$ and all $s_{k_1}, s_{k_2} \in \mathcal{S} = \{s_1, ..., s_K\} \subset \mathbb{N}_0$.*
**Generalized empirical notions of group fairness** In practice, exact equality according to corollary 2.6 is usually not achieved. This motivates keeping the difference between both sides of the equation(s) as small as possible, which translates to the following two *generalized* definitions of disparate impact and demographic parity (DP).

More precisely, while for group fairness, the majority of the literature focuses on a binary classification model $\hat{Y}$ and a single binary sensitive feature $S$ (cf. *Feldman et al., 2015*; *Hardt, Price & Srebro, 2016*; *Kamiran & Calders, 2009*; *Kamiran & Calders, 2010*; *Mehrabi et al., 2021*; *Pessach & Shmueli, 2022*; *Ruf & Detyniecki, 2021*; *Zafar et al., 2017a, 2017b*), in this work's definitions, we generalize the understanding of group fairness to a non-binary sensitive feature $S$, but which can also be used to model even multiple non-binary sensitive features (remark 2.9).

While disparate impact is specifically designed for a binary classification task, *i.e.*, for a setting where $\mathcal{Y} = \{0, 1\}$ holds, and where the class $\{Y = 1\}$ is the preferred one (remark 2.11), the demographic parity score additionally allows generalization to a one- or multidimensional non-binary classifier $\hat{Y}$ by definition and based on the theoretical background in subsection "Independence of two random variables"[7]:

**Definition 2.7** (Disparate impact).

Let $\epsilon \in [0, 1]$. The disparate impact score

$$\text{DI} := \min_{s_{k_1}, s_{k_2} \in \mathcal{S}} \frac{\mathbb{P}(\hat{Y} = 1 \mid S = s_{k_1})}{\mathbb{P}(\hat{Y} = 1 \mid S = s_{k_2})}$$

measures the (un-)fairness of the classification model $\hat{Y}$ with respect to the sensitive feature $S$ in the sense of the independence criterion. For the model $\hat{Y}$, disparate impact is limited to $\epsilon$ iff $\text{DI} \geq 1 - \epsilon$ holds.

**Definition 2.8** (Demographic parity).

Let $\epsilon \in [0, 1]$. The demographic parity score

$$\text{DP} := \max_{y \in \mathcal{Y}, \, s_{k_1}, s_{k_2} \in \mathcal{S}} \left| \mathbb{P}(\hat{Y} = y \mid S = s_{k_1}) - \mathbb{P}(\hat{Y} = y \mid S = s_{k_2}) \right|$$

measures the (un-)fairness of the classification model $\hat{Y}$ with respect to the sensitive feature $S$ in the sense of the independence criterion. For the model $\hat{Y}$, demographic parity holds with respect to $\epsilon$ iff $\text{DP} \leq \epsilon$ holds.

*Remark* 2.9 In our previous work (cf. *Strotherm & Hammer, 2023*), we consider $K = |\mathcal{S}|$ different binary random variables $S_1, ..., S_K : \Omega \to \{0, 1\}$ when defining disparate impact. Encoding the single non-binary random variable $S$ from this work to $K$ such binary random variables $S_1, ..., S_K$ for all $k = 1, ..., K$ yields the same definition of disparate impact as given in *Strotherm & Hammer (2023)*. We change the notation in this work because it is more intuitive compared to common fairness definitions (*e.g.*, cf. Eq. (2.2) and proof of lemma 2.10) and easily shows how these fairness definitions can be extended even to *multiple* non-binary sensitive features: In this case, the random *vector* $S = (S_1, ..., S_{d_s}) : \Omega \to \mathcal{S}$ with $\mathcal{S} = \mathcal{S}_1 \times ... \times \mathcal{S}_{d_s} \subset \mathbb{N}_0^{d_s}$ and $d_s > 1$ encodes all $d_s$ possibly non-binary single sensitive features $S_l$ for $l = 1, ..., d_s$.

---

[7] If $\mathcal{Y} \neq \{0, 1\}$ is discrete, testing for the canonical one-elementary events $y \in \mathcal{Y}$ is still a necessary and sufficient condition for independence according to lemma 2.2.

**Accordance of empirical notions of group fairness in the binary case** In case of a binary classification task and a single binary sensitive feature, our definitions coincide with the according definitions known from the before-mentioned literature:

**Lemma 2.10** *If $\mathcal{Y} = \mathcal{S} = \{0, 1\}$ holds, the disparate impact score DI and the demographic parity score DP according to definition 2.7 and 2.8, respectively, coincide with the corresponding definitions known from the literature.*

*Proof.* If $\mathcal{Y} = \mathcal{S} = \{0, 1\}$ holds, the fact that $\{\hat{Y} = 0\} \dot\cup \{\hat{Y} = 1\} = \Omega$ holds implies that the probability measure $\mathbb{P}(\hat{Y} \in \cdot \mid S = s)$ is uniquely determined by the probability $\mathbb{P}(\hat{Y} = 1 \mid S = s)$ for all $s \in \mathcal{S}$. Therefore, the independence criterion (corollary 2.6) becomes

$$\mathbb{P}(\hat{Y} = 1 \mid S = 0) = \mathbb{P}(\hat{Y} = 1 \mid S = 1).$$

By the same fact,

$$\begin{aligned} &\left| \mathbb{P}(\hat{Y} = 0 \mid S = 0) - \mathbb{P}(\hat{Y} = 0 \mid S = 1) \right| \\ =\ & \left| 1 - \mathbb{P}(\hat{Y} = 1 \mid S = 0) - (1 - \mathbb{P}(\hat{Y} = 1 \mid S = 1)) \right| \\ =\ & \left| \mathbb{P}(\hat{Y} = 1 \mid S = 0) - \mathbb{P}(\hat{Y} = 1 \mid S = 1) \right| \end{aligned}$$

holds. Therefore, the demographic parity score (definition 2.8) becomes

$$\mathrm{DP} = \left| \mathbb{P}(\hat{Y} = 1 \mid S = 0) - \mathbb{P}(\hat{Y} = 1 \mid S = 1) \right|.$$

Moreover, the disparate impact score (definition 2.7) becomes

$$\mathrm{DI} = \min \left\{ \frac{\mathbb{P}(\hat{Y} = 1 \mid S = 0)}{\mathbb{P}(\hat{Y} = 1 \mid S = 1)}, \frac{\mathbb{P}(\hat{Y} = 1 \mid S = 1)}{\mathbb{P}(\hat{Y} = 1 \mid S = 0)} \right\}.$$

In most literature, where $\{S = 0\}$ is assumed to be the discriminated group, and therefore, where $\mathbb{P}(\hat{Y} = 1 \mid S = 0) < \mathbb{P}(\hat{Y} = 1 \mid S = 1)$ holds, this simplifies to

$$\mathrm{DI} = \frac{\mathbb{P}(\hat{Y} = 1 \mid S = 0)}{\mathbb{P}(\hat{Y} = 1 \mid S = 1)}$$

(cf. Eq. (2.2)). These are the definitions of the disparate impact and the demographic parity score usually found in the literature (cf. *Mehrabi et al., 2021*; *Pessach & Shmueli, 2022*; *Ruf & Detyniecki, 2021*; *Zafar et al., 2017b*).

As already briefly touched on in the subsection "Motivation", the disparate impact criterion assures that the relative amount of positive predictions within the discriminated group $\{S = 0\}$ – or in our generalized case of non-binary sensitive features, within the most discriminated group—deviates at most $(100 - p)\% = 100\epsilon\%$ from the relative amount of positive predictions within the privileged group $\{S = 1\}$—or in our generalized case, within the most privileged group (definition 2.7). For short and in either way: It aims at obtaining similar or equal *success* or *passing rates* among groups.

Similarly, in a binary classification task, the demographic parity criterion assures that the relative amount of positive predictions deviates at most $100\epsilon\%$ among groups (cf. proof

of lemma 2.10 or Table 2). In contrast, in a non-binary classification task, the demographic parity criterion assures that the relative amount of *any* predictions deviates at most $100\epsilon\%$ among groups (definition 2.8).

By that, while both criteria assure similar or equal passing rates among groups in the setting of a binary classification task, they assure different things in the setting of a non-binary classification task due to the consideration of all labels in the demographic parity criterion (Table 2).

*Remark* 2.11 (Generalizability of the disparate impact score). Similar to the demographic parity score DP (definition 2.8), one could ask whether it makes sense to generalize the disparate impact score DI to arbitrary discrete target spaces $\mathcal{Y}$ by

$$\text{DI}^* := \min_{y \in \mathcal{Y}, \, s_{k_1}, s_{k_2} \in \mathcal{S}} \frac{\mathbb{P}(\hat{Y} = y \mid S = s_{k_1})}{\mathbb{P}(\hat{Y} = y \mid S = s_{k_2})}. \tag{2.3}$$

However, this generalized definition would *not* coincide with the common one from Eq. (2.2) in the setting of lemma 2.10: For example, consider the case

$$\mathbb{P}(\hat{Y} = 1 \mid S = 0) = 0.8, \quad \mathbb{P}(\hat{Y} = 1 \mid S = 1) = 0.9,$$
$$\mathbb{P}(\hat{Y} = 0 \mid S = 0) = 0.2, \quad \mathbb{P}(\hat{Y} = 0 \mid S = 1) = 0.1.$$

In this case, the disparate impact score according to definition 2.7 is equal to $DI = \min\{\frac{0.8}{0.9}, \frac{0.9}{0.8}\} = \frac{0.8}{0.9} \approx 0.88$, which usually is a score considered to be fair. In contrast, the disparate impact score according to Eq. (2.3) is equal to $DI^* = \min\{DI, \frac{0.2}{0.1}, \frac{0.1}{0.2}\} = \frac{0.1}{0.2} = 0.5$, which usually is a score considered to be unfair. The reason is that the idea of disparate impact relies on the fact that the class $\{Y = 1\}$ is the desired one and only the relative amount of positive predictions among groups is of interest (cf. *Pessach & Shmueli, 2022*). Therefore, it does only make sense to define disparate impact score as we do in definition 2.7.

### Separation

**An easy-to-test notion of group fairness** Depending on the application, one disadvantage of fairness notions that belong to the fairness concept *independence* could be the missing dependence on the true label $Y$. In such case, even if the model $\hat{Y}$ was perfect, *i.e.*, if $\hat{Y} = Y$ held, it would be denoted as unfair if the relative amount of positive training labels differed significantly among groups (cf. *Hardt, Price & Srebro, 2016*).

The solution to that yields the fairness concept *separation*, which in contrast to the fairness concept independence requires (mathematical) independence between the model's prediction $\hat{Y}$ and the sensitive feature $S$, conditioned on $Y$:

**Definition 2.12** (Fairness according to the separation criterion (cf. *Barocas, Hardt & Narayanan, 2019*)).

The classification model $\hat{Y}$ is fair with respect to the sensitive feature $S$ in the sense of the separation criterion iff $\hat{Y}$ and $S$ are mathematically independent with respect to $\mathbb{P}(\cdot \mid Y = y)$ for all $y \in \mathcal{Y}$.

Using the modified probability measure $\mathbb{P}(\cdot \mid Y = y)$ for $y \in \mathcal{Y}$, lemma 2.2 again induces the following easy-to-test separation criterion in the context of fair ML:

**Corollary 2.13** (Fairness according to the separation criterion).

*The classification model $\hat{Y}$ is fair with respect to the sensitive feature $S$ in the sense of the separation criterion iff*

$$\mathbb{P}(\hat{Y} = \hat{y} \mid S = s_{k_1}, Y = y) = \mathbb{P}(\hat{Y} = \hat{y} \mid S = s_{k_2}, Y = y)$$

*holds for all $y, \hat{y} \in \mathcal{Y} = \{0, 1\} \subset \mathbb{N}_0$ and all $s_{k_1}, s_{k_2} \in \mathcal{S} = \{s_1, ..., s_K\} \subset \mathbb{N}_0$.*

**Generalized empirical notions of group fairness** Again, in practice, exact equality according to corollary 2.13 is usually not achieved. Therefore, again, keeping the difference between both sides of the equation as small as possible motivates the following *generalized* definitions, where similar to the previous subsection "Independence", the second one is specifically designed for a binary classification task, *i.e.*, for settings where $\mathcal{Y} = \{0, 1\}$ holds, and where the class $\{Y = 1\}$ is the preferred one.

**Definition 2.14** (Equalized odds).

Let $\epsilon \in [0, 1]$. The equalized odds scores

$$EOs(y) := \max_{s_{k_1}, s_{k_2} \in \mathcal{S}} \left| \mathbb{P}(\hat{Y} = y \mid S = s_{k_1}, Y = y) - \mathbb{P}(\hat{Y} = y \mid S = s_{k_2}, Y = y) \right|$$

measure the (un-)fairness of the classification model $\hat{Y}$ with respect to the sensitive feature $S$ in the sense of the separation criterion for all $y \in \mathcal{Y}$. For the model $\hat{Y}$, equalized odds hold with respect to $\epsilon$ iff $EOs(y) \leq \epsilon$ holds for all $y \in \mathcal{Y}$.[8]

**Definition 2.15** (Equal opportunity).

Let $\epsilon \in [0, 1]$. The equal opportunity (EO) score

$$EO := \max_{s_{k_1}, s_{k_2} \in \mathcal{S}} \left| \mathbb{P}(\hat{Y} = 1 \mid S = s_{k_1}, Y = 1) - \mathbb{P}(\hat{Y} = 1 \mid S = s_{k_2}, Y = 1) \right|$$

measures the (un-)fairness of the classification model $\hat{Y}$ with respect to the sensitive feature $S$ in the sense of the separation criterion. For the model $\hat{Y}$, equal opportunity holds with respect to $\epsilon$ iff $EO \leq \epsilon$ holds.

Similar arguments as compared to subsection "Independence" also show how these definition(s) allow a generalized understanding of group fairness for non-binary and even multiple non-binary sensitive features $S$, and for a one- or multi-dimensional non-binary classifier $\hat{Y}$.

**Accordance of empirical notions of group fairness in the binary case** In case of a binary classification task and a single binary sensitive feature, our definitions coincide with the according definitions known from other literature:

**Lemma 2.16.** *If $\mathcal{Y} = \mathcal{S} = \{0, 1\}$ holds, the equalized odds scores $EOs(y)$ for $y \in \mathcal{Y}$ and the equal opportunity score EO according to definition 2.14 and 2.15, respectively, coincide with the corresponding definitions known from the literature.*

*Proof.* If $\mathcal{Y} = \mathcal{S} = \{0, 1\}$ holds, similar to the proof of lemma 2.10, the separation criterion (definition 2.13) becomes

[8] Instead of the requirement "for all $y \in \mathcal{Y}$", we could also take the maximum over $y \in \mathcal{Y}$ and $s_{k_1}, s_{k_2} \in \mathcal{S}$ and consider a single equalized odds score as done in definition 2.8.

$$\mathbb{P}(\hat{Y} = 1 \mid S = 0, Y = y) = \mathbb{P}(\hat{Y} = 1 \mid S = 1, Y = y)$$

for $y = 0, 1$, and

$$
\begin{aligned}
&\left|\mathbb{P}(\hat{Y} = 0 \mid S = 0, Y = 0) - \mathbb{P}(\hat{Y} = 0 \mid S = 1, Y = 0)\right| \\
&= \left|1 - \mathbb{P}(\hat{Y} = 1 \mid S = 0, Y = 0) - (1 - \mathbb{P}(\hat{Y} = 1 \mid S = 1, Y = 0))\right| \\
&= \left|\mathbb{P}(\hat{Y} = 1 \mid S = 0, Y = 0) - \mathbb{P}(\hat{Y} = 1 \mid S = 1, Y = 0)\right|
\end{aligned}
$$

holds.[9] Therefore, the equalized odds scores (definition 2.14) become

$$EOs(1) = \left|\mathbb{P}(\hat{Y} = 1 \mid S = 0, Y = 1) - \mathbb{P}(\hat{Y} = 1 \mid S = 1, Y = 1)\right| \text{ and}$$

$$EOs(0) = \left|\mathbb{P}(\hat{Y} = 1 \mid S = 0, Y = 0) - \mathbb{P}(\hat{Y} = 1 \mid S = 1, Y = 0)\right|$$

(comparison of true positive rates (TPRs) and false positive rates (FPRs) among groups) and the equal opportunity score (definition 2.15) becomes

$$EO = \left|\mathbb{P}(\hat{Y} = 1 \mid S = 0, Y = 1) - \mathbb{P}(\hat{Y} = 1 \mid S = 1, Y = 1)\right|$$

(comparison of TPRs among groups). These are the definitions of the equalized odds and the equal opportunity score(s) usually found in the literature (cf. *Mehrabi et al., 2021*; *Pessach & Shmueli, 2022*; *Ruf & Detyniecki, 2021*; *Zafar et al., 2017a*).

While equalized odds ensure that the true positive rates (TPRs) *and* true negative rates (TNRs) (or equivalently, false positive rates (FPRs)) among groups differ at most $100\epsilon\%$ in a binary classification task, equal opportunity only concentrates on TPRs among groups. In contrast, in a non-binary classification task where the TPRs and FPRs are not well-defined, equalized odds refer to similar or equal *correct classification rates* per label among groups (cf. definition 2.14) and display a natural generalization of equal opportunity in this setting (cf. definition 2.15).

*Remark* 2.17. Nevertheless, we will not make use of equalized odds in this work, as the TNRs and FPRs given by $\mathbb{P}(\hat{Y} = 0 \mid S = s, Y = 0)$ and $\mathbb{P}(\hat{Y} = 1 \mid S = s, Y = 0)$ for any $s \in \mathcal{S}$, respectively, do not exist in our domain of application although being in the setting of a binary classification task, as we will see in subsection "Fairness in leakage detection".

### Summary of generalized notions of group fairness

To conclude, in this section, we derived generalized exact and empirical notions of group fairness based on the mathematical concept of independence and suitable for a single, but also multiple non-binary sensitive feature(s). All exact and some empirical notions are suitable for not only one-, but also multi-dimensional non-binary classification models. We additionally showed that the notions coincide with common group fairness definitions in the case of a binary classification task and a single binary sensitive feature.

A summary of such already existing definitions and our contributions are summarized in Tables 1 and 2.

*Remark* 2.18 (Computation of group fairness scores in practice). In practice, the true distributions $\mathbb{P}(S, \hat{Y})^{-1}$ and $\mathbb{P}(S, Y, \hat{Y})^{-1}$, on which the probabilities displayed in Tables 1 and 2 are based, are unknown. Therefore, as elaborated in subsection "Mathematical

[9] More precisely, comparing the TNRs is equivalent to comparing the FPRs.

notation for machine learning", the fairness scores are computed using the empirical approximations $\widehat{\mathbb{P}}(S, \hat{Y})^{-1} = \frac{1}{n}\sum_{i=1}^{n} \delta_{(s_i, \hat{y}_i)}$ and $\widehat{\mathbb{P}}(S, Y, \hat{Y})^{-1} = \frac{1}{n}\sum_{i=1}^{n} \delta_{(s_i, y_i, \hat{y}_i)}$ based on the training data $\mathcal{D}$, respectively, yielding the required approximated probabilities

$$\widehat{\mathbb{P}}(\hat{Y} = y \mid S = s) = \frac{\sum_{i=1}^{n} \mathbb{1}_{\{\hat{y}_i = y, s_i = s\}}}{\sum_{i=1}^{n} \mathbb{1}_{\{s_i = s\}}} \quad \text{and}$$

$$\mathbb{P}(\hat{Y} = \hat{y} \mid S = s, Y = y) = \frac{\sum_{i=1}^{n} \mathbb{1}_{\{\hat{y}_i = \hat{y}, s_i = s, y_i = y\}}}{\sum_{i=1}^{n} \mathbb{1}_{\{s_i = s, y_i = y\}}} \quad \text{for all } y, \hat{y} \in \mathcal{Y}, s \in \mathcal{S}.$$

# LEAKAGE DETECTION IN WATER DISTRIBUTION SYSTEMS

In view of the AI Act, by being part of the critical infrastructure, WDSs belong to *high-risk systems* (cf. *Veale & Borgesius, 2021*). In this context, "(m)uch attention has been paid to the potential for AI systems to facilitate indirect discrimination, (which is) in principle illegal under EU law" (cf. *Veale & Borgesius, 2021*). One requirement of such systems is therefore to check the system for bias and to document the system's performance for different demographic groups (cf. *Strotherm et al., 2023*). While this could suggest the use of group fairness definitions implicitly, the guidelines for trustworthy AI explicitly name fairness as one of the seven essential requirements for such systems (cf. *European Union, 2019*).

A key challenge in the domain of WDSs where AI, or more precisely, ML, is used, is to detect leakages (cf. *Artelt et al., 2022*; *Guo et al., 2021*; *Li et al., 2022*; *Romero-Ben et al., 2022*; *Steffelbauer et al., 2022*; *Vrachimis et al., 2022*). The main components of a WDS relevant for this work are *nodes* and *pipes*, through which water can be supplied to end users such as private households, hospitals or schools located at the nodes of the network, but which are also vulnerable to leakages. To detect these is therefore crucial to guarantee consistent water supply, but can also affect other important tasks such as short-term decision making and long-term planning of WDSs.

Therefore, as requested by the AI Act and the guidelines for trustworthy AI, in this section, we present a common ML-based pipeline and concrete instantiations of how to detect leakages in WDSs (subsection "Methodology of leakage detection"). Consecutively, we investigate what fairness can mean (subsection "Fairness in leakage detection") and whether it is satisfied (subsections "Application domain and data set and experimental results" and "Analysis: Residual-based ensemble leakage detection does not obey fairness") in this context according to common group fairness notions as introduced in subsection "Generalized notions of group fairness in machine learning".

## Methodology of leakage detection

In the task of leakage detection, the domain $\Omega$ (cf. subsection "Mathematical notation for machine learning") corresponds to possible states of a WDS, determined by

time-dependent demands of the end users located at the $D \in \mathbb{N}$ nodes in the system and which may be affected by leakages. We assume that among these nodes, $d \in \mathbb{N}$ nodes are provided with sensors (usually, $D \gg d$), which deliver pressure measurements $p(t) \in \mathbb{R}^d$ at different times $t \in \mathbb{R}$ and which can be used for the task at hand. As the sensors usually measure pressure values within fixed time intervals $\delta \in \mathbb{R}_+$, we introduce the notation $t_i := t_0 + i\delta$, where $t_0$ is some fixed reference point with respect to time.

There are several methodologies that make use of such pressure measurements to approach the problem of leakage detection using ML. Using the notation from subsection "Mathematical notation for machine learning", the goal is to train a binary classifier $\hat{Y} = f(X) : \Omega \to \mathcal{X} \to \mathcal{Y}$ with $\mathcal{Y} = \{0, 1\}$ that predicts the true state $Y : \Omega \to \mathcal{Y}$ of the WDS with respect to the question whether a leakage is active ($\{Y = 1\}$) or not ($\{Y = 0\}$). Hereby, the feature space $\mathcal{X}$ depends on the specific method but is related to the before-mentioned pressure measurements.

One standard approach comes in three steps (cf. *Isermann, 2006*): First of all, so called *virtual sensors* are trained, *i.e.*, regression models that are able to predict the pressure at some time $t_i \in \mathbb{R}$ and at a node $j \in \{1, \ldots, d\}$ (or even $j \in \{1, \ldots, D\}$), based on the pressure measurements observed at that (or earlier) time(s) and at (a choice of) the sensor nodes $j \in \{1, \ldots, d\}$. Subsequently, these virtual sensors are used to compute *pressure residuals* of measured and predicted pressure. Finally, these pressure residuals are fed into a *leakage detector* $\hat{Y}$ that is able to predict whether a leakage is present in the WDS at the time of the used residual (cf. *Isermann, 2006*). An overview of this pipeline is displayed in Fig. 1.

The approach can differ depending on the concrete instantiation of virtual sensors and the leakage detector. In this subsection, we first formalize the idea of the general leakage detection pipeline described above in more detail (subsection "Leakage detection pipeline"). Consecutively, we present two concrete instantiations of such (subsection "Leakage detection instantiations"), which we will investigate with respect to the question of fairness in the rest of this section.

### Leakage detection pipeline
**Virtual sensors** Based on vector inputs $\tilde{p}^j(t_i) \in \mathbb{R}^{d_r}$ that are based on the pressure measurements $p(t_i) = (p_j(t_i))_{j=1,\ldots,d} \in \mathbb{R}^d$ observed at (multiple) times $t_i \in \mathbb{R}$ and at the sensor nodes $j = 1, \ldots, d$ in the WDS, so called *virtual sensors*, *i.e.*, regression models

$$f_j^r : \mathbb{R}^{d_r} \to \mathbb{R}$$
$$\tilde{p}^j(t_i) \mapsto f_j^r(\tilde{p}^j(t_i))$$

that predict the pressure at times $t_i \in \mathbb{R}$ and at the sensor node $j$ are trained for each sensor node $j = 1, \ldots, d$. Hereby, the dimension $d_r \in \mathbb{N}$ and the inputs $\tilde{p}^j(t_i) \in \mathbb{R}^{d_r}$ depend on the specific model architecture used (cf. subsection "Leakage detection instantiations" and *Artelt et al., 2022*; *Ashraf et al., 2023*; *Isermann, 2006*).

**Pressure residuals** Independent of what specific instantiations of virtual sensors $f_j^r$ for $j = 1, \ldots, d$ are used, standard leakage detection methods rely on the *pressure residuals*

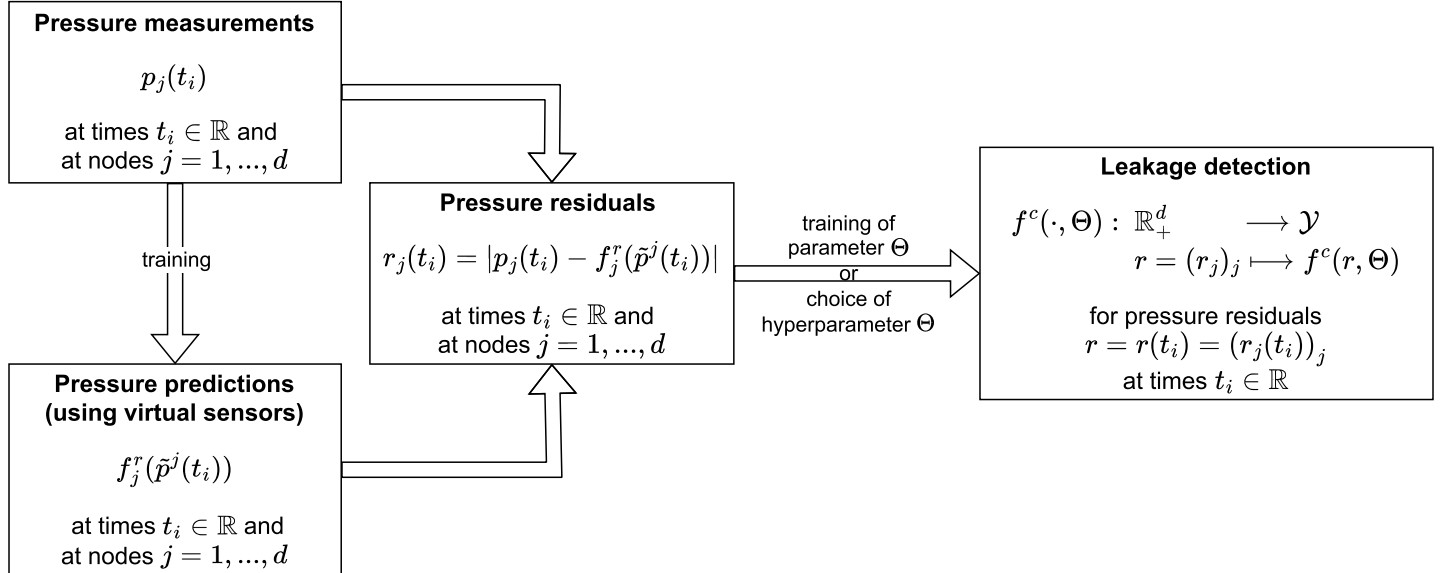

**Figure 1   Standard leakage detection pipeline.**         

$$r_j(t_i) := |p_j(t_i) - f_j^r(\tilde{p}^j(t_i))| \in \mathbb{R}_+$$

we obtain from the pressure measurements $p_j(t_i) \in \mathbb{R}$ and the pressure predictions $f_j^r(\tilde{p}^j(t_i)) \in \mathbb{R}$ at (possibly unseen) times $t_i \in \mathbb{R}$ and at the sensor node $j$ for all $j = 1, \ldots, d$ (cf. *Artelt et al., 2022*; *Isermann, 2006*).

**Leakage detection** Based on pressure residuals $r(t_i) = (r_j(t_i))_{j=1,\ldots,d} \in \mathcal{X} := \mathbb{R}^{d_c} = \mathbb{R}^d$ (*i.e.*, $d_c = d$) at times $t_i \in \mathbb{R}$ and at the sensor nodes $j = 1, \ldots, d$ in the WDS, a classification model $\hat{Y}$—or more precisely, the learnable model $f^c$ which is applied to the feature pressure residuals $X : \Omega \to \mathcal{X}$ (cf. subsection "Mathematical notation for machine learning").

$$f^c := f_\Theta^c := f^c(\cdot, \Theta) : \mathcal{X} \to \mathcal{Y}$$

that predicts whether ($\{\hat{Y} = f^c(X) = 1\}$) or not ($\{\hat{Y} = f^c(X) = 0\}$) a leakage is present in the WDS is defined or trained. Hereby, $\Theta \in \mathbb{R}^{d_p}$ indicates a choosable or trainable (hyper-) parameter and the hypothesis space $\mathcal{H}$ depends on the specific model architecture used (subsections "Leakage detection instantiations", "Fairness-enhancing leakage detection in water distribution networks" and *Artelt et al., 2022*; *Isermann, 2006*).

### Leakage detection instantiations

The previous subsection gives a general pipeline on how to detect leakages in a WDS based on pressure measurements, pressure predictions based on virtual sensors, resulting pressure residuals and finally, the leakage detection itself (cf. Fig. 1). In this subsection, we present specific instantiations of this approach.

**Linear virtual sensors** The first approach is based on the work of *Artelt et al. (2022)*: In this case, each virtual sensor $f_j^r : \mathbb{R}^{d_r} \to \mathbb{R}$ at each sensor node $j \in \{1, \ldots, d\}$ corresponds

to a linear regression model. The inputs $\tilde{p}^j(t_i) \in \mathbb{R}^{d_r}$ at times $t_i \in \mathbb{R}$ consist of rolling means

$$\tilde{p}^j(t_i) := \bar{p}_{\neq j}(t_i) := \frac{1}{T_r + 1} \sum_{\iota=0}^{T_r} p_{\neq j}(t_i - \iota\delta) \in \mathbb{R}^{d-1}$$

at all sensor nodes except the node $j$ and with a to be chosen time window $T_r + 1 \in \mathbb{N}$. By that, each regression model's input dimension equals $d_r := d - 1$.

Based on that, the $d$ virtual sensors $f_j^r$ for each sensor node $j \in \{1, ..., d\}$ are trained on leakage free training data $\mathcal{D}_j^r = \{(\bar{p}_{\neq j}(t_i), p_j(t_i)) \in \mathbb{R}^{d_r} \times \mathbb{R} \mid i = 0, ..., n_r\}$. More precisely, $y(t_i) = 0 \in \mathcal{Y}$ holds for all realisations $i = 0, ..., n_r$ of the label $Y$.

**GCN virtual sensors** In contrast, the second approach is based on the work of *Ashraf et al. (2023)*: In this case, each virtual sensor $f_j^r : \mathbb{R}^{d_r} \to \mathbb{R}$ at each sensor node $j \in \{1, ..., d\}$ is obtained by training a single GCN model.

The GCN model is trained on leakage free training data

$$\mathcal{D}^r = \{((p_j(t_i))_{j=1,...,d}, (p_j(t_i))_{j=1,...,D}) \in \mathbb{R}^d \times \mathbb{R}^d \mid i = 0, ..., n_r\}.$$

More precisely, the GCN model inputs the sparse pressure measurements at the sensor nodes $j = 1, ..., d$ and outputs the pressure predictions at *each* node $j = 1, ..., D$ of the WDS. However, for this work, the pressure predictions at the sensor nodes $j = 1, ..., d$ are enough: The $d$ virtual sensors $f_j^r$ at each sensor node $j \in \{1, ..., d\}$ can be considered as the entry-wise output of the overall GCN model $f^r := (f_j^r)_{j=1,...,D}$.

By that, the inputs $\tilde{p}^j(t_i) \in \mathbb{R}^{d_r}$ at times $t_i \in \mathbb{R}$ are given by the node-independent pressure measurements $\tilde{p}(t_i) = \tilde{p}^j(t_i) := (p_{\hat{j}}(t_i))_{\hat{j}=1,...,d} \in \mathbb{R}^d$ themselves for *all* sensor nodes $j = 1, ...d$, and each regression model's input dimension equals $d_r := d$.

**Ensemble leakage detection: The H-method** Independent on the choice of virtual sensor, based on the pressure residuals $r(t_i) = (r_j(t_i))_{j=1,...,d} \in \mathcal{X} = \mathbb{R}_+^{d_c} = \mathbb{R}_+^d$ at times $t_i \in \mathbb{R}$ we obtain from these, a simple leakage detection method performing good on standard benchmarks, is the *threshold-based ensemble classification* introduced by *Artelt et al. (2022)*: Without any further training, we can choose a node-dependent hyperparameter $\theta_j \in \mathbb{R}_+$ to define a (local) classifier $f_j^c : \mathbb{R}_+ \to \mathcal{Y}$ for each sensor node $j \in \{1, ...d\}$ by

$$f_j^c(r_j(t_i)) = f_j^c(r_j(t_i), \theta_j) := \mathbb{1}_{\{r_j(t_i) > \theta_j\}}.$$

We then obtain an ensemble classifier $f^c : \mathcal{X} \to \mathcal{Y}$ with feature space $\mathcal{X} = \mathbb{R}_+^{d_c} = \mathbb{R}_+^d$ and hyperparameter $\Theta := (\theta_j)_{j=1,...,d_c} \in \mathcal{X}$ (i.e., $d_p = d_c = d$) that predicts whether there is a leakage present in the WDS at time $t_i \in \mathbb{R}$ or not, defined by

$$f^c(r(t_i)) = f^c(r(t_i), \Theta) := \mathbb{1}_{\left\{\sum_{j=1}^{d_c} f_j^c(r_j(t_i)) \geq 1\right\}}. \tag{3.4}$$

Simply put into words, a node-dependent classifier $f_j^c$ detects a leakage when the node-dependent pressure-residual $r_j(t_i) \in \mathbb{R}_+$ at time $t_i \in \mathbb{R}$ exceeds the node-dependent threshold $\theta_j \in \mathbb{R}_+$ and the ensemble classifier $f^c$ detects a leakage when one of the node-dependent classifiers $f_j^c$ for any $j \in \{1, ..., d\}$ does.

We call this overall instantiation of the standard leakage detection pipeline (cf. Fig. 1) independent of the instantiation of the virtual sensors and characterized by choosing the **H**yperparameter $\Theta \in \mathcal{X}$ the **H-method**. Note that the H-method does not need further training once it has access to feature pressure residuals $\mathcal{X} = \mathbb{R}_+^d$. How to introduce a trainable structure to this last component of the pipeline will be part of subsection "Fairness-enhancing leakage detection in water distribution networks".

### Fairness in leakage detection

After having introduced a pipeline to define a leakage detection model $\hat{Y} = f^c(X)$ and possible concrete instantiations of such in the previous subsection, the question arises as to how leakage detection is related to fairness in the sense of subsection "Generalized notions of group fairness in machine learning". One key contribution of this work is to answer this question, *i.e.*, to introduce the notion of fairness in the application domain of WDSs by defining suitable sensitive features in the context of leakage detection or other ML-based services in WDSs.

**Sensitive features in ML-based services in WDS** Knowing that each node of the WDS corresponds to a group of consumers, a natural question is whether these local groups benefit from the WDS and its related services, such as leakage detection, in equal degree. To ensure that the methods that will be presented in subsection "Methodology of fairness-enhancing leakage detection" scale to larger WDSs, we do not consider single nodes but $K \in \mathbb{N}$ groups of nodes in the WDS as protected groups in terms of fairness. Then, given that a leakage is active in the WDS, *i.e.*, that $Y = 1$ holds, we define the sensitive feature $S \in \mathcal{S} := \{s_1, ..., s_K\} := \{1, ..., K\}$ to answer the question where, *i.e.*, in which protected group $k \in \{1, ..., K\}$, this leakage is active.[10] In terms of equal service, one would expect an equally good detection of leakages independent on the leakage location, *i.e.*, the protected group. This understanding of sensitive features, protected groups and consecutively, fairness in WDS, can of course be adapted to other ML-based services in WDS, for example, to contamination detection.

**Fairness notions in ML-based services in WDS** In this work, we will focus on the evaluation of fairness by choosing one fairness notion each from the fairness concepts independence and separation (subsections "Independence" and "Separation"): Disparate impact for independence (definition 2.7) due to its importance also in the legal context (cf. subsection "Motivation"), and equal opportunity for separation (definition 2.15). Regarding the latter concept, considering that our sensitive feature $S$ is defined on the event $\{Y = 1\}$, this shows why using equalized odds is not possible in this setting, as already mentioned in remark 2.17 and as shown in the proof of the next lemma 3.1.

**Fairness properties in ML-based services in WDS** Given *this* definition of a non-binary sensitive feature $S$ in the WDS, we obtain the following important results with regard to the notions of fairness chosen.

**Lemma 3.1** (Equivalence of disparate impact and equal opportunity in WDSs). *Let* $S : \{Y = 1\} \to \{1, ..., K\}$ *be the sensitive feature describing where a leakage in one of the protected groups* $k \in \{1, ..., K\}$ *of the WDS is active. Moreover, let* $\epsilon, \tilde{\epsilon} \in [0, 1]$ *and define* $\max_k := \max_{k \in \{1, ..., K\}} \mathbb{P}(\hat{Y} = 1 \mid S = k).$

[10] Note that in contrast to common settings, where the random variables $Y, \hat{Y}$ and $S$ share the same domain, with respect to this fairness question, for the sensitive feature $S$, we change the domain from the population space $\Omega$ of all possible states to those, in which a leakage is present, *i.e.*, to $\{Y = 1\} \subset \Omega$.

*1. If disparate impact is limited to $\epsilon$, equal opportunity holds with respect to $\tilde{\epsilon} = \epsilon \max_k$.*

*2. If equal opportunity holds with respect to $\tilde{\epsilon}$, disparate impact is limited to* $\epsilon = \tilde{\epsilon}(\max_k)^{-1}$.

*Proof.* First of all, note that for any $\omega \in \Omega$ for which there exists a $k \in \{1, \ldots, K\}$ such that $S(\omega) = s_k = k$ holds, $Y(\omega) = 1$ must hold by definition of the sensitive feature $S$ (this is why it only makes sense to define $S$ on $\{Y = 1\} \subset \Omega$). Therefore, $\{S = k, Y = 0\}$ is empty for all $k = 1, \ldots, K$. Subsequently, we obtain

$$\begin{aligned}
\{S = k, Y = 1\} &= \{S = k, Y = 1\} \cup \{S = k, Y = 0\} \\
&= \{S = k\} \cap (\{Y = 1\} \cup \{Y = 0\}) \\
&= \{S = k\} \cap \Omega \\
&= \{S = k\}
\end{aligned}$$

and thus, $\mathbb{P}(\hat{Y} = 1 \mid S = k, Y = 1) = \mathbb{P}(\hat{Y} = 1 \mid S = k)$ for all $k = 1, \ldots, K$.

Secondly, we also define $\min_k := \min_{k \in \{1, \ldots, K\}} \mathbb{P}(\hat{Y} = 1 \mid S = k)$. Then, we easily find that $DI = \frac{\min_k}{\max_k}$ and, together with the first observation, $EO = \max_k - \min_k$ holds (cf. definition 2.7 and 2.15).

Now the rest follows by simple equivalent transformations: In setting 1, we find that

$$\frac{\min_k}{\max_k} \geq 1 - \epsilon \Leftrightarrow \min_k \geq (1 - \epsilon)\max_k \Leftrightarrow \max_k - \min_k \leq \epsilon \max_k \tag{3.5}$$

holds. In setting 2, we obtain

$$\max_k - \min_k \leq \tilde{\epsilon} \Leftrightarrow 1 - \frac{\min_k}{\max_k} \leq \frac{\tilde{\epsilon}}{\max_k} \Leftrightarrow \frac{\min_k}{\max_k} \geq 1 - \frac{\tilde{\epsilon}}{\max_k}. \tag{3.6}$$

**Corollary 3.2.** *Given the setting of lemma 3.1,*

*1. $EO = \widetilde{EO}$ for $\widetilde{EO} := (1 - DI) \cdot \max_k$ and*

*2. $DI = \widetilde{DI}$ for $\widetilde{DI} := 1 - \frac{EO}{\max_k}$ holds.*

*Proof.* This is a direct consequence of lemma 3.1, where we choose $\epsilon := 1 - DI$ in setting 1 and $\tilde{\epsilon} := EO$ in setting 2, and where we can work with equalities instead of estimations in Eqs. (3.5) and (3.6), respectively.

## Application domain and data set

After having introduced an appropriate definition of a sensitive feature and protected groups in WDSs in the previous subsection "Fairness in leakage detection", in order to test whether the concrete instantiations of leakage detection methods presented in subsection "Leakage detection instantiations" are fair in this sense, we need to generate suitable data based on given WDS structures.

**The WDSs** considered are *Hanoi* (cf. *Santos-Ruiz et al., 2022*; *Vrachimis et al., 2018*) and *L-Town* (cf. *Vrachimis et al., 2022*; *Vrachimis et al., 2020*) displayed in Figs. 2 and 3, respectively. While Hanoi consists of 32 nodes, among which three are provided with sensors, and 34 links, L-Town displays a more realistic WDS consisting of 785 nodes,
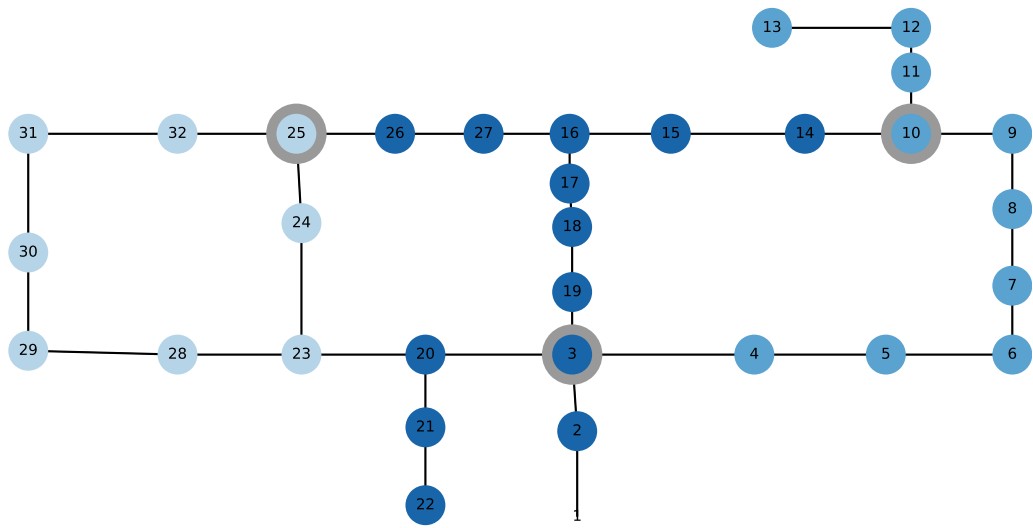

**Figure 2 The Hanoi WDS, its sensor nodes (IDs 3, 10 and 25) and the protected groups, each highlighted in another color (group 1 on the left side in light shade, group 2 in the middle in dark shade, group 3 on the right side in middle shade).** The sensor nodes are colored in the same color as the protected group to which they belong and highlighted with a grey circle.

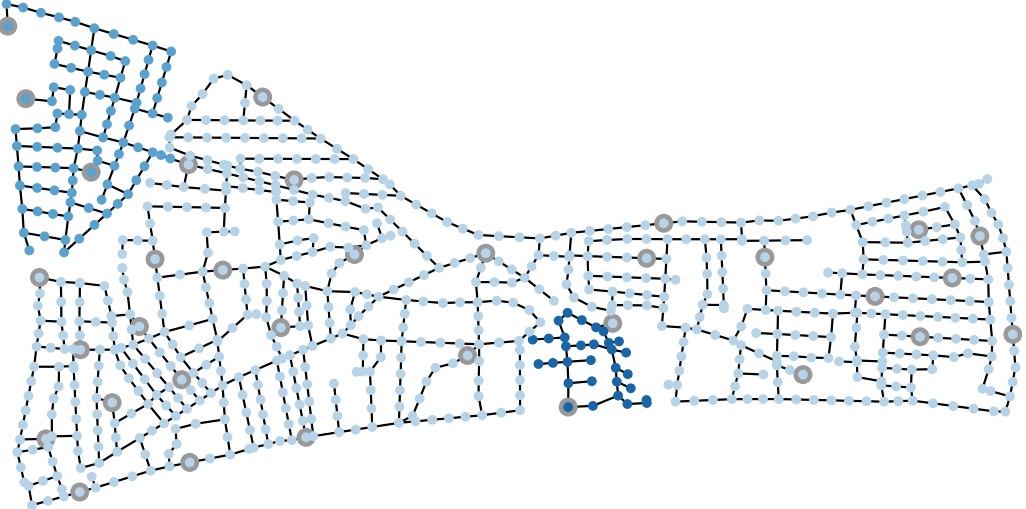

**Figure 3 The L-Town WDS, its sensor nodes and the protected groups, each highlighted in another color (group 1, also called area C, on the top left in middle shade; group 2, also called area A, in light shade; group 3, also called area B, on the bottom in the middle in dark shade).** The sensor nodes are colored in the same color as the protected group to which they belong and highlighted with a grey circle.

among which 33 are provided with sensors, and 909 links. The latter is constructed in a way to mimic a true WDS while satisfying security defaults and displays one of the state-of-the-art WDSs in the water domain.

**Pressure measurement simulation** For security reasons, only a limited number of real-world data sets based on such systems are available. Therefore, to evaluate methods

**Table 3** **Results of the H-method.** Results of the H-method with $\max_k$ and $\min_k$ according to (the proof of) lemma 3.1. Moreover, the disparate impact and equal opportunity score $DI$ and $EO$ as well as $\widetilde{DI}$ and $\widetilde{EO}$ according to definition 2.7, 2.15, corollary 3.2.2 and 3.2.1, respectively.

| d | ACC | $\max_k$ | $\min_k$ | DI | EO | $\widetilde{DI}$ | $\widetilde{EO}$ |
|---|---|---|---|---|---|---|---|
| 5 | 0.6223 | 0.8468 | 0.4880 | 0.5763 | 0.3558 | 0.5763 | 0.3588 |
| 10 | 0.7998 | 0.9983 | 0.6372 | 0.6383 | 0.3611 | 0.6383 | 0.3611 |
| 15 | 0.8837 | 1.0000 | 0.6402 | 0.6402 | 0.3598 | 0.6402 | 0.3598 |
| (a) Hanoi WDS and linear virtual sensors. | | | | | | | |
| 1.9 | 0.7034 | 0.8935 | 0.4828 | 0.5404 | 0.4107 | 0.5404 | 0.4107 |
| 2.3 | 0.8346 | 1.0000 | 0.6652 | 0.6652 | 0.3348 | 0.6652 | 0.3348 |
| 2.7 | 0.8476 | 1.0000 | 0.4254 | 0.4254 | 0.5746 | 0.4254 | 0.5746 |
| (b) L-Town WDS and GCN virtual sensors. | | | | | | | |

such as the H-method presented in subsection "Leakage detection instantiations", data has to be simulated.

For Hanoi, we generate pressure measurements with a time window of $\delta = 10$ min. using the atmn toolbox (cf. *Vaquet et al., 2023*). The pressure is simulated at the sensor nodes displayed in Fig. 2 and for different leakage scenarios, which differ in the leakage location and size. As the WDS is relatively small, we are able to simulate a leakage at each node in the system and for three different diameters $d \in \{5, 10, 15\}$ cm. In total, the data set is balanced with respect to the label, *i.e.*, the fact whether ($\{Y = 1\}$) or not ($\{Y = 0\}$) a leakage present at the time of the considered sample.

For L-Town, we generate pressure measurements with a time window of $\delta = 5$ min. as used in the work of *Ashraf et al. (2023)*. The pressure is simulated at the sensor nodes displayed in Fig. 3 and for different leakage scenarios. Due to the larger system size, we are only able to simulate a leakage at some nodes in the system and for three different diameters $d \in \{1.9, 2.3, 2.7\}$ cm[11].

**Pressure residual computation** Consecutively, in order to obtain the pressure residuals required for the H- or other method(s), virtual sensor predictions have to be generated (Fig. 1). For Hanoi, we train and use linear virtual sensors with a preprocessing hyperparameter of $T_r = 2$ as done by *Artelt et al. (2022)*. For L-Town, we train and use GCN virtual sensors, (cf. subsection "Leakage detection instantiations").

**Protected groups** Finally, the protected groups as introduced in subsection "Fairness in leakage detection" are displayed in Figs. 2 and 3 as well. Here, we work with $K = 3$ different groups for both the Hanoi and the L-Town WDS.

## Experimental results and analysis: Residual-based ensemble leakage detection does not obey fairness

In Table 3, the results of the H-method presented in subsection "Leakage detection instantiations" are shown. The hyperparameter $\Theta \in \mathcal{X} = \mathbb{R}_+^d$ is chosen manually per diameter d such that the test accuracy is close to maximal. On the one hand, we see that independent of the WDS and the virtual sensors used, in general, the larger the leakage

[11] The different leakage sizes for the two WDSs Hanoi and L-Town can be explained based on some physical background onto which we can not comment in full detail. However, roughly speaking, the size of the chosen leakage diameters depend on the water supply and demand dynamics of a WDS. For Hanoi, the number of consumers is small, however, water demands are high since the water source has sufficient water pressure. Hence, larger leakage diameters (proportional to demands) are required to simulate a significant leak. For L-Town, the number of consumers is much larger that has smaller individual demands since the water pressure at the reservoirs is the same as Hanoi. Hence, smaller leakage diameters are sufficient to simulate significant leakages.

size, the better the method performs in terms of accuracy (ACC), as larger leakages are associated with larger pressure drops. Moreover, the method is capable of detecting even small leakages with high(er) accuracy in larger (and therefore, more realistic) WDSs (cf. footnote 11 for details).

On the other hand, and more importantly, we see that independent of the WDS and the virtual sensors used, the method is unfair in terms of disparate impact score *DI*, where a value of 0.8 or larger is desirable (cf. *Zafar et al., 2017b*), and equal opportunity score *EO*. However, the experimental evaluation confirms the mathematical findings of corollary 3.2 by comparing the column of the disparate impact score calculated according to definition 2.7 (*DI*) to the one according to corollary 3.2.2 ($\widetilde{DI}$), and the column of the equal opportunity score calculated according to definition 2.15 (*EO*) to the one according to corollary 3.2.1 ($\widetilde{EO}$). This also justifies that in our setting, the usage of one of the two scores is sufficient. Therefore, from now on, we mostly work with the disparate impact score *DI* only.

## FAIRNESS-ENHANCING LEAKAGE DETECTION IN WATER DISTRIBUTION NETWORKS

Motivated by the result that the standard leakage detection method presented in subsection "Leakage detection instantiations" does not satisfy the notions of fairness, as another main contribution of this work, we modify this H-method to enhance fairness as introduced in subsection "Generalized notions of group fairness in machine learning". The main idea is based on the fact that in the H-method the only models trained are the virtual sensors $f_j^r$ for all $j = 1, \ldots, d$ (cf. subsection "Leakage detection instantiations"). However, given these virtual sensors and resulting residuals $r(t_i) \in \mathcal{X} = \mathbb{R}_+^d$, as well as labels $y(t_i) \in \mathcal{Y} = \{0, 1\}$ for times $t_i \in \mathbb{R}$, we can turn the choice of the hyperparameter $\Theta := (\theta_j)_{j=1,\ldots,d} \in \mathcal{X}$ of the ensemble classifier $f^c = f^c(\cdot, \Theta)$ (cf. Eq. (3.4)) into an optimization problem (OP), where $\Theta \in \mathcal{X}$ now acts as a parameter. The corresponding hypothesis space is $\mathcal{H} := \{f^c : \mathcal{X} \to \mathcal{Y}, \ r \mapsto f^c(r, \Theta) \mid \Theta \in \mathcal{X}\}$ (cf. subsection "Mathematical notation for machine learning").

In the following section, we therefore propose (subsection "Methodology of fairness-enhancing leakage detection") and evaluate (subsection "Experimental results and analysis") different, in contrast to the H-method optimization-based, methods that aim at optimizing the parameter $\Theta \in \mathcal{X}$ in order to obtain an optimal ensemble classifier $f^c(\cdot, \Theta_{\text{opt.}}) \in \mathcal{H}$. Optimality hereby depends on the OP at hand: These methods on the one hand are further baselines, where treating the modeling problem as an OP enables us to optimize the result of the H-method itself without fairness considerations. On the other hand, we consider fairness-enhancing methods, where the parameter $\Theta \in \mathcal{X}$ needs to be optimized such that the resulting ensemble classifier is simultaneously as accurate and fair on the given training data as possible.

### Methodology of fairness-enhancing leakage detection

The following methods define training algorithms to find an optimal ensemble classifier $f^c(\cdot, \Theta_{\text{opt.}}) \in \mathcal{H}$. The scores considered in these algorithms rely on labeled training data

[12] In practice, we train and test the (ensemble) classifier(s) on unseen data for times $i \geq n_r + 1$. However, for the sake of readability, we choose the indices $i = 1, \ldots, n_c$ instead of $i = n_r + 1, \ldots, n_c$.

$\mathcal{D}^c = \{(r(t_i), y(t_i)) \in \mathcal{X} \times \mathcal{Y} \mid i = 1, \ldots, n_c\}$[12], which also holds samples based on leaky states $\{Y = 1\}$ of the WDS. For simplicity, we omit the dependence of all considered functions on the training data $\mathcal{D}^c$.

### Fair leakage detection framework

In general, a learning problem such as the training of an optimal ensemble classifier $f^c(\cdot, \Theta_{\text{opt.}}) \in \mathcal{H}$ can be phrased as an OP, where the objective is to minimize some suitable loss function $L = L(\Theta) := L(f^c(\cdot, \Theta), \cdot) : \mathcal{X} \times \mathcal{Y} \to \mathbb{R}$ over the hypothesis space $\mathcal{H}$, or more precisely, with respect to the parameter $\Theta \in \mathcal{X}$, based on its evaluations on the training data $\mathcal{D}^c \subset \mathcal{X} \times \mathcal{Y}$:

$$\min_{\Theta \in \mathcal{X}} L(\Theta). \tag{4.7}$$

The advantage of redefining the choice of hyperparameters $\Theta \in \mathcal{X}$ (which is what we do in the H-method) as an OP is that we can now extend this OP by side constraints $C_k = C_k(\Theta) := C_k(f^c(\cdot, \Theta), \cdot) : \mathcal{X} \times \mathcal{Y} \to \mathbb{R}$:

$$\begin{cases} \min_{\Theta \in \mathcal{X}} & L(\Theta), \\ \text{s.t.} & C_k(\Theta) \geq 0 \ \forall k = 1, \ldots, \hat{K}. \end{cases} \tag{4.8}$$

A typical way of optimizing a constrained OP is to integrate the side constraints in the objective in order to apply unconstrained optimization algorithms. This can be done using a *barrier-* or *penalty function* $p : \mathbb{R} \to [-\infty, \infty]$ (cf. *Nocedal & Wright, 2006*). Using such functions, the constrained OP Eq. (4.8)[13] can be transformed to

[13] Note that the requirement $\Theta = (\theta_j)_{j=1,\ldots,d_c} \in \mathcal{X} = \mathbb{R}_+^d$ actually also contains the constraint $\theta_j \geq 0$ for all $j = 1, \ldots, d$. Nevertheless, also the residuals $r(t_i) \in \mathcal{X} = \mathbb{R}_+^d$ are non-negative for all $i = 1, \ldots, n$. Therefore, if for any $j \in \{1, \ldots, d\}$, $\theta_j < 0$ holds, the ensemble classifier $\hat{Y} = f^c(X, \Theta)$ (Eq. (3.4)) is equal to $\hat{Y} \equiv 1$, *i.e.*, it *only* predicts leakages. As our datasets are balanced with respect to the labels, this will lead to an accuracy of approximately 0.5 and to a TPR, but also an FPR of 1. Thus, such choices are no (local) optima of the OPs as they either do not deliver a(n) (locally) optimal loss or as they harm the side constraint(s). In other words, the solution of the OPs will automatically be feasible with respect to the constraint $\theta_j \geq 0$ for all $j = 1, \ldots, d$. Therefore, for simplicity, we do not include this constraint as a regularization term in the objective and can optimize $\Theta$ over $\mathbb{R}^d$ instead of $\mathcal{X} = \mathbb{R}_+^d$ anyways.

$$\begin{cases} \min_{\Theta \in \mathcal{X}} & L(\Theta) + \mu \sum_{k=1}^{\hat{K}} p(C_k(\Theta)). \end{cases} \tag{4.9}$$

Hereby, the hyperparameter $\mu \in [0, \infty)$ regulates the importance of the constraints $C_k$ for all $k = 1, \ldots, \hat{K}$ compared to the loss function $L$.

### Fair leakage detection instantiations

Equation (4.9) gives a general framework on how to train a (fair) leakage detection model based on the general leakage detection pipeline presented in subsection "Leakage detection pipeline". While the H-method presented in subsection "Leakage detection instantiations" is an instantiation of this pipeline that only requires the training of the virtual sensors, *i.e.*, the first component of the pipeline, the following methods also require the training of the leakage detection model itself, *i.e.*, the third component of the pipeline (cf. Fig. 1).

More precisely, the following methods are instantiations of this third component using the framework proposed in the previous subsection "Fair leakage detection framework". Hereby, the ensemble classifier $f^c(\cdot, \Theta) \in \mathcal{H}$ on which the loss function $L$ and the side constraints $C_k$ for $k = 1, \ldots, \hat{K}$ rely is of the same structure as in the H-method (cf. Eq. (3.4)); the resulting optimal models $\hat{Y} = f^c(X, \Theta_{\text{opt.}})$ only differ in their optimal parameter $\Theta_{\text{opt.}} \in \mathcal{X} = \mathbb{R}_+^d$.

We obtain such different optimal parameters by choosing different loss functions $L$, different side constraints $C_k$ for $k = 1, \ldots, \hat{K}$ and different algorithmic choices. In the following, we propose such possible choices. The indices (loss index, constraint index,

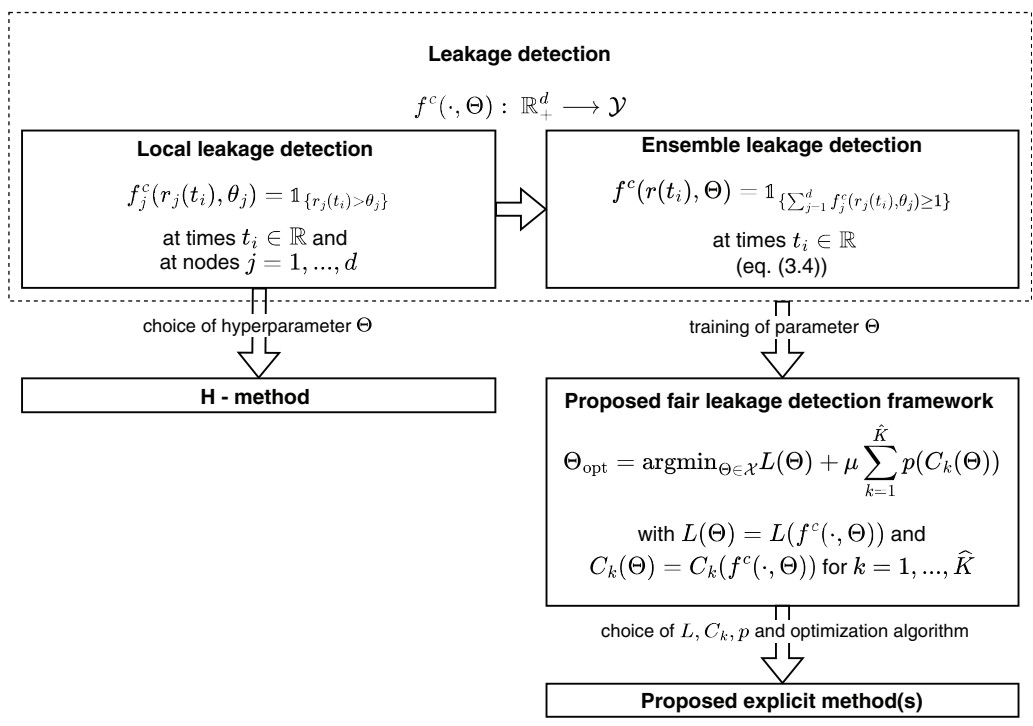

**Figure 4** **Fair leakage detection framework as an extension of Fig. 1.**

optimization index and barrier or penalty function index) introduced along the way will later be used for the names of the resulting explicit methods as combinations of such choices. A general scheme of this overall idea as an extension of Fig. 1 is displayed in Fig. 4.

**Optimizing performance as baseline methods** By choosing a typical evaluation score as the loss function $L$ and not using any further (fairness) constraints (*i.e.*, $\mu = 0$ or $\hat{K} = 0$), we obtain further baseline methods which output optimized parameters $\Theta_{\text{opt.}} = \text{argmin}_{\Theta \in \mathcal{X}} L(\Theta)$ compared to the H-method and by that, with respect to the performance of the leakage detection model $\hat{Y} = f^c(X, \Theta_{\text{opt.}})$, but not with respect to its fairness.

Typical such evaluation scores for a binary classification task are:

- The negative accuracy, *i.e.* $L(\Theta) = -\text{ACC}(\Theta)$ (loss index "ACC"),
- the negative difference $L(\Theta) = -\text{TPR}(\Theta) + \text{FPR}(\Theta)$ between the TPR and the FPR (loss index "TFPR").

**Optimizing Performance under Fairness Constraints** For the following approaches, the loss function $L$ controls the performance while the constraints $C_k$ control the fairness for all $k = 1, ..., \hat{K}$.

*Choice of performance loss functions:* When optimizing the performance under fairness constraints, we choose the same loss functions as when optimizing the performance without fairness constraints as introduced in the previous paragraph.

[14] As discussed in subsection "Application domain and data set", while the model $\hat{Y}$ is defined on the population space $\Omega$, the sensitive feature $S$ is defined on $\{Y = 1\} \subset \Omega$. Moreover, the empirical covariance is only well-defined for variables that are metric scaled or binary nominal scaled.

[15] More precisely, $\{S = k\} = \{S_k = 1\}$ holds, with the advantage that the model $\hat{Y}$ and the binary sensitive features $S_k$ for all $k = 1, \ldots, K$ are defined on the same domain; on the population space $\Omega$.

*Choice of fairness constraints:* As done in our previous work (cf. *Strotherm & Hammer, 2023*), in terms of fairness constraints, we make use of the covariance between the sensitive feature(s) and the prediction of the ensemble classifier. For technical reasons[14], we have to transform the non-binary sensitive feature $S : \{Y = 1\} \to \{1, \ldots, K\}$ to $K$ binary sensitive features $S_k : \Omega \to \{0, 1\}$, which gives answer to the question of whether ($\{S_k = 1\}$) or not ($\{S_k = 0\}$) a leakage is active in group $k$ for all $k = 1, \ldots, K$.[15] Using that $\hat{y}(t_i) = f^c(r(t_i), \Theta)$ holds for all realisations $i = 1, \ldots, n_c$, for all binary sensitive features $S_k$ for $k = 1, \ldots, K$, the *empirical* covariance between a single sensitive feature $S_k$ and the model $\hat{Y} = f^c(X, \Theta)$ is given by

$$\text{Cov}_{\text{emp.}}(S_k, \hat{Y}) = n_c^{-1} \sum_{i=1}^{n_c} (s_k(t_i) - \overline{s_k}) \cdot f^c(r(t_i), \Theta). \tag{4.10}$$

The usage of the (empirical) covariance as a proxy for fairness is based on the idea that group fairness of a model $\hat{Y}$, or more precisely, a high disparate impact score on which we focus in this work, relies on the assumption of $\hat{Y}$ being independent of the sensitive feature $S$ (cf. subsection "Independence"), or in our case, each of the sensitive features $S_k$ for $k = 1, \ldots, K$. As the independence of two random variables implies their covariance being equal to zero, the latter can be interpreted as a necessary condition for fairness. For more information on this intuition, but also on how our contributions are generalizations of the work of *Zafar et al. (2017b)*, we refer to our previous work *Strotherm & Hammer (2023)*.

Motivated by that, we require $\text{Cov}_{\text{emp.}}(S_k, \hat{Y}) \le c$ and $\text{Cov}_{\text{emp.}}(S_k, \hat{Y}) \ge -c$ to hold, or, equivalently formulated in standard form:

- We require $C_k(\Theta) := c - \text{Cov}_{\text{emp.}}(S_k, \hat{Y}) \ge 0$ and $C_k(\Theta) := c + \text{Cov}_{\text{emp.}}(S_k, \hat{Y}) \ge 0$ to hold for all $k = 1, \ldots, K$ (*i.e.*, $\hat{K} = 2K$ in Eq. (4.8)). Hereby, the hyperparameter $c \in [0, \infty)$ regulates how much the covariance's absolute value is bounded and therefore, the desired fairness (constraint index "COV").

**Optimizing Fairness under Performance Constraints** For the following approaches, the loss function $L$ controls the fairness while the constraints $C_k$ control the performance for all $k = 1, \ldots, \hat{K}$.

*Choice of fairness loss functions:* As done in our previous work *Strotherm & Hammer (2023)*, we choose the disparate impact score DI as a loss function. Moreover, as elaborated in the conclusion of our previous work *Strotherm & Hammer (2023)* and similar to *Zafar et al. (2017b)*, we additionally change the role of the empirical covariance by optimizing a fairness proxy similar to the one introduced in Eq. (4.10) directly. Therefore, taking into account that we have multiple sensitive values, two reasonable loss functions are:

- The sum $L(\Theta) := \sum_{k=1}^{K} |\text{Cov}(S_k, \hat{Y})|$ of absolute values of the empirical covariance between a single sensitive feature $S_k$ and the model $\hat{Y} = f^c(X, \Theta)$ for all $k = 1, \ldots, K$ (loss index "Cov"),

- the negative disparate impact score $L(\Theta) := -DI(\Theta)$ (definition 2.7) (loss index "DI").

*Choice of performance constraints:* In terms of performance constraints, we stick to the choice of the accuracy ACC, which is only allowed to differ by some percentage of the optimal accuracy $ACC_{opt.}$ obtained when training without fairness constraints (cf. *Strotherm & Hammer, 2023*; *Zafar et al., 2017b*). More precisely, we require $ACC(\Theta) \geq (1 - \lambda)ACC_{opt.}$ or, equivalently formulated in standard form:

- We require $C_1(\Theta) := ACC(\Theta) - (1 - \lambda)ACC_{opt.} \geq 0$ to hold (*i.e.*, $\hat{K} = 1$ in Eq. (4.8)). Hereby, the hyperparameter $\lambda \in [0, 1]$ regulates how much the accuracy $ACC(\Theta)$ is allowed to differ from the optimal accuracy $ACC_{opt.}$ received, *e.g.*, by another baseline method (constraint index "ACC").

By that, it indirectly regulates the fairness as well, as the more the accuracy is allowed to differ from the optimal accuracy, the larger the feasible subspace of $\mathcal{X}$ gets and by that, the more the fairness as the loss in the objective can be optimized.

**Algorithmic choices** Next to the choices of loss function and constraints, the final methods also differ in dependence of what algorithmic choices are made, *e.g.*, what optimization algorithm as well as what barrier or penalty function $p$ is used (cf. Eq. (4.9)).

One question to answer when choosing an optimization algorithm is whether the considered objective of the OP is (continuously) differentiable. In the setting of ML, the objective clearly depends on the model's prediction $\hat{Y} = f^c(X, \Theta)$ or more precisely, on $y(t_i) = f^c(r(t_i), \Theta)$ for all $i = 1, \dots, n_c$. However, in view of the ensemble classifier's definition (cf. Eq. (3.4)), $f^c$ is not differentiable with respect to $\Theta$.

Therefore, in dependence on the fact whether we chose a differentiable (db) or non-differentiable (ndb) optimization algorithm, we need to approximate the model:

- If we want to use a gradient-based optimization technique, we make $\hat{Y} = f^c(X, \cdot)$ differentiable by approximating each indicator function $\mathbb{1}_{\{v > 0\}}$ by the sigmoid function $\text{sgd}_b(v) = (1 + \exp^{-bv})^{-1}$ with hyperparameter $b \in [0, \infty)$ (optimization index "db"). Replacing the ensemble classifier's prediction $f^c(r(t_i), \Theta)$ (cf. Eq. (3.4)) by

$$\hat{f}^c(r(t_i), \Theta) := \text{sgd}_b\left(\sum_{j=1}^{d_c} \text{sgd}_b(r_j(t_i) - \theta_j) - T\right)$$

for all $i = 1, \dots, n_c$ yields a differentiable approximation of the model $\hat{Y}$. Hereby, we replace the threshold 1 of the exact ensemble classifier $f^c$ with a hyperparameter $T \in [0, 1]$ to handle the insecurity of the sigmoid function around zero.

- If we want to use a non gradient-based optimization technique, we do not make any changes (optimization index "ndb").

For more details on that, we refer to our previous work *Strotherm & Hammer (2023)*. Depending on what optimization algorithm is used, different (differentiable or non-differentiable) barrier or penalty functions $p$ can be used. In this work, we make use of

- the barrier function $p(t) := -\log(t)$ (barrier function index "log") and
- the penalty function $p(t) := \max\{0, -t\}$ (penalty function index "max").

**Explicit methods** Finally, after having presented all possible choices, we obtain the following explicit methods using the following nomenclature:

loss index+[constraint index–optimization index–barrier/penalty function index].

Each resulting fairness-enhancing method comes with a corresponding baseline method to which it will be compared in the evaluation:

- the fairness-enhancing TFPR+COV-db-log-method with corresponding baseline TFPR-db-method,
- the fairness-enhancing TFPR+COV-ndb-log- and TFPR+COV-ndb-max-method with corresponding baseline TFPR-ndb-method,
- the fairness-enhancing ACC+COV-db-log-method with corresponding baseline ACC-db-method,
- the fairness-enhancing ACC+COV-ndb-log- and ACC+COV-ndb-max-method with corresponding baseline ACC-ndb-method,
- the fairness-enhancing COV+ACC-ndb-log- and COV+ACC-ndb-max-method also with corresponding baseline ACC-ndb-method and
- the fairness-enhancing DI+ACC-ndb-log- and DI+ACC-ndb-max-method also with corresponding baseline ACC-ndb-method.

The first two notes refer to the fairness-enhancing methods where performance is optimized under fairness constraints and the last four notes refer to the fairness-enhancing methods where fairness is optimized under performance constraints.

## Experimental results and analysis

Based on the pressure measurements in the Hanoi WDS and the pressure residuals we obtain from these measurements by making use of the virtual sensors (cf. subsection "Application domain and data set"), we test all methods introduced in subsections "Leakage detection instantiations" (H-method) and "Fair leakage detection instantiations in practice". Afterwards, we will test the best performing method on the data associated with the more complex and more realistic L-Town WDS.

*Training and testing setup:* To test the considered methods, a model $\hat{Y}$ is trained per method and per leakage diameter d on training data (40% of the overall data) and evaluated on test data (60% of the overall data).[16] For the training, the different OPs presented in subsection "Methodology of fairness-enhancing leakage detection" are solved using the BFGS algorithm (cf. *Nocedal & Wright, 2006*) in case of a differentiable OP and the Downhill-Simplex-Search algorithm, also known as the Nelder-Mead algorithm, (cf. *Gao & Han, 2012*) in case of a non-differentiable OP in order to find the optimal parameter $\Theta_{\text{opt.}} \in \mathcal{X}$ of the leakage detection model $\hat{Y} = f^c(X, \Theta_{\text{opt.}})$.

The implementation of all methods and all our results can be found on GitHub (https://github.com/jstrotherm/FairnessInWDSs_extended).

[16] Note that since for the fairness evaluation, enough data from all different groups is required, we choose a comparably high percentage of the data for testing.

### Hanoi

**Initial parameters** Optimization algorithms require an initial start parameter. For the experiments on the Hanoi WDS, we use the hyperparameter $\Theta_{opt.} \in \mathcal{X}$ found for the H-method (cf. subsection "Methodology of leakage detection") as an initial parameter $\Theta_0 \in \mathcal{X}$ for the remaining optimization-based methods (cf. subsection "Methodology of fairness-enhancing leakage detection").

**Hyperparameters** While the parameters $\Theta_{opt.} \in \mathcal{X}$ are now outputs of these optimization-based methods, these are subordinate to other hyperparameters. In Table 4, an overview of these hyperparameters are displayed per method (and if required, per diameter d). We choose suitable hyperparameters $\mu, b \in [0, \infty)$ and $T \in [0, 1]$ and keep them fixed afterwards. In contrast, the fairness-hyperparameters $c \in [0, \infty)$ or $\lambda \in [0, 1]$, *i.e.*, the hyperparameters that regulate the fairness directly or indirectly, respectively, are changed to obtain different *score combinations* of performance, measured by the accuracy score *ACC*, and fairness, measured by the disparate impact or equal opportunity score *DI* or *EO*, respectively. We do so by starting with a hyperparameter $c$ or $\lambda$ that causes perfect fairness, *i.e.*, a disparate impact score of 1.0, whenever possible and in- or decrease the hyperparameter by 0.01 until the disparate impact score of the resulting fairness-enhanced model achieves an equal or worse disparate impact score than its corresponding baseline method, respectively (cf. paragraph "Explicit methods" in subsection "Fair leakage detection instantiations" or Table 4 for the corresponding baseline method per fairness-enhancing method).

**Results** With these settings in mind, we obtain the following results. As we in total test five baseline methods (the H-method and the ones proposed in subsection "Fair leakage detection instantiations") and 10 fairness-enhancing methods (cf. subsection "Fair leakage detection instantiations"), and by that, many methods, we only present the key findings in this section and further detailed findings regarding the comparison of all these methods in Appendix B.

Moreover, for a better overview of the results, we divide the ten fairness-enhancing methods into four subcategories: The TFPR-methods including all methods with loss index "TFPR", and analogously the ACC-methods, the COV-methods and the DI-methods.

In some of the results, these methods are represented together with their corresponding baseline methods. Note that two methods from the same subcategory can have different baseline methods as corresponding baseline methods (cf. paragraph "Explicit methods" in subsection "Fair leakage detection instantiations" or Table 4).

*Increasing fairness:* In Fig. 5, we see the performance and fairness of some exemplary trained ensemble classifiers measured in accuracy and disparate impact score, respectively. For the fairness-enhancing methods, testing different hyperparameters $c$ or $\lambda$ cause error bars for these methods. The height of the bars with error bars corresponds to the mean accuracy and disparate impact score achieved by each method over all hyperparameter values tested. The error bars themselves reach from the lowest to the largest score of the two scores considered.

**Table 4 Overview of the used hyperparameters per method and possibly per diameter d.** The "b" indicates baseline methods that aim at optimizing general performance without fairness considerations. For more details on these hyperparameters, see subsection "Fair leakage detection instations".

| Method | $c \in [0, \infty)$ | $\lambda \in [0, 1]$ | $\mu$ (d = 5, 10, 15) | $b$ | $T$ |
|---|---|---|---|---|---|
| TFPR-db (b) | – | – | – | 100 | 0.8 |
| TFPR+COV-db-log | ✓ | – | 0.10 0.20 0.20 | 100 | 0.8 |
| TFPR-ndb (b) | – | – | – | – | – |
| TFPR+COV-ndb-log | ✓ | – | 0.20 0.25 0.25 | – | – |
| TFPR+COV-ndb-max | ✓ | – | 100 | – | – |
| ACC-db (b) | – | – | – | 100 | 0.8 |
| ACC+COV-db-log | ✓ | – | 0.15 0.05 0.05 | 100 | 0.8 |
| ACC-ndb (b) | – | – | – | – | – |
| ACC+COV-ndb-log | ✓ | – | 0.2 0.3 0.05 | – | – |
| ACC+COV-ndb-max | ✓ | – | 100 | – | – |
| COV+ACC-ndb-log | – | ✓ | 0.01 0.01 0.01 | – | – |
| COV+ACC-ndb-max | – | ✓ | 100 | – | – |
| DI+ACC-ndb-log | – | ✓ | 0.05 0.025 0.04 | – | – |
| DI+ACC-ndb-max | – | ✓ | 100 | – | – |

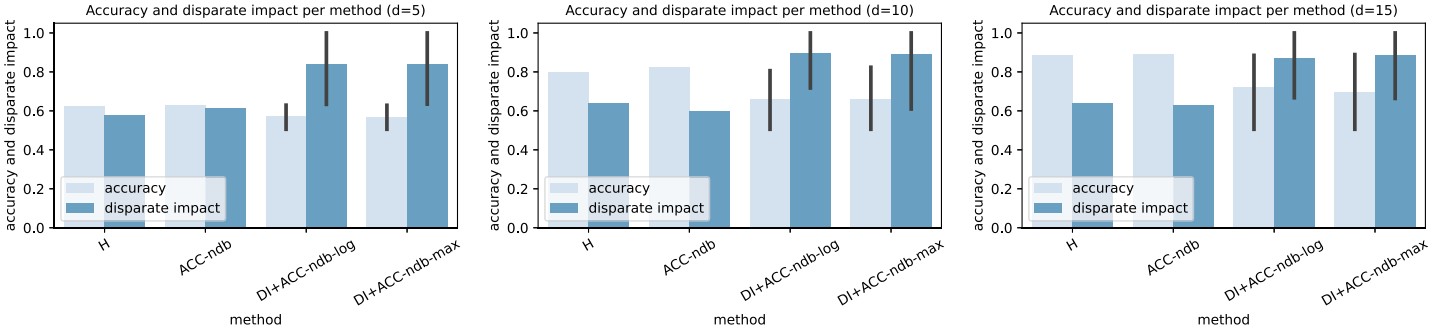

**Figure 5 Accuracy and disparate impact score per method and leakage diameter in the Hanoi-WDS as well as for different hyperparameters $c$ or $\lambda$.**

We see that the fairness-enhancing methods on average increase fairness while on average decrease accuracy compared to their corresponding baseline methods. However, the average increase in fairness is larger than the average decrease in accuracy. For details regarding different diameters d, the score ranges and the other methods, we refer to Appendix B. Based on these, one can say that fairness and overall performance are mutually dependent to about the same extent.

In addition to that, Fig. 6 shows the performance and indirectly, also the fairness of some exemplary trained ensemble classifiers measured by the TPR per group. The height of the bars and the range of the error bars behave analogously to Fig. 5.

In view of the definition of the equal opportunity score (cf. definition 2.15) and due to the fact that this score is equivalent to the disparat impact score in our domain of

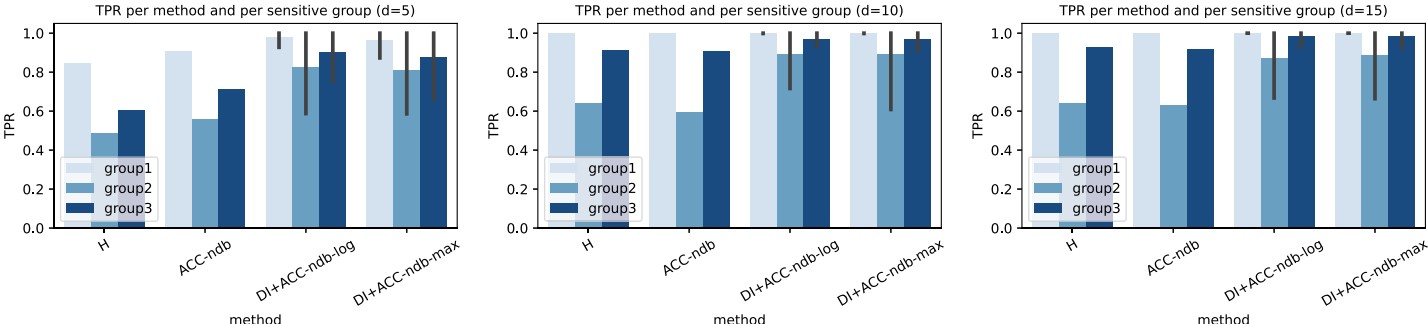

**Figure 6 TPR per method, group and leakage diameter in the Hanoi-WDS as well as for different hyperparameters $c$ or $\lambda$.**

application (cf. lemma 3.1 and corollary 3.2), in our context, the more similar the TPRs per group are, the fairer a method is. This is what we observe in Fig. 6 (and Fig. B.2) when comparing the TPRs among groups for the fairness-enhancing methods to the TPRs among groups for their corresponding baseline methods. Even more, Fig. 6 (and Fig. B.2) show(s) that the increase in fairness that we observe in Fig. 5 (and Fig. B.1) on average is not only obtained by decreasing the performance of the (in the corresponding baseline method) best-performing group but also by increasing the performance of the (in the corresponding baseline method) worst-performing group. By some methods, even all TPRs per group are increased on average.

*The coherence of fairness and overall performance, and non-optimality:* While Figs. 5 and 6 only hint at the relationship between fairness and overall performance, measured in disparate impact and accuracy score, respectively, a more detailed visualization of how fairness is related to the overall performance of a model can be found in Fig. 7. For each tested hyperparameter $c$ or $\lambda$, respectively, depending on what fairness-enhancing method was used, the obtained score combinations, *i.e.*, the accuracy and the disparate impact score, are visualized for some exemplary trained ensemble classifiers.

The characteristic curve that can be observed in most of all sub-images is called the pareto-front, visualizing that the increase in fairness is accompanied by the reduction in accuracy score and vice versa. Note that the non-optimal solutions apart from the pareto-front in Fig. 7 and also later on, the local jumps recognized in Fig. 8, can be explained by the non-convexity of the objective functions. Because of that, the found solutions $\Theta_{\text{opt.}} \in \mathcal{X}$ strongly depend on the initialized parameter $\Theta_0 \in \mathcal{X}$ and might not correspond to the global optimum.

Nevertheless, by most fairness-enhancing methods, a desired disparate impact score of about 0.8 can be achieved by a decrease of accuracy by approximately 0.03–0.06 points below the optimal accuracy obtained by the corresponding baseline methods (depending on the specific method used). Hereby, both fairness and overall performance can be influenced by the fairness-hyperparameters $c$ or $\lambda$, respectively. Deciding which choice of fairness-hyperparameter is optimal and by that, deciding on the trade-off between fairness and overall performance, is a difficult task that depends on the extent of the decisions of

the underlying model as well as legal requirements. Regarding legal requirements, by not using the sensitive features for the decision making of the algorithms, the methods presented can satisfy the legal definition of disparate treatment *and* disparate impact (depending on the hyperparameter chosen) simultaneously.

Another observation is that the largest accuracies of the fairness-enhancing methods are usually approximately as good as the accuracy of their corresponding baseline methods while achieving equal or better fairness results. In the opposite direction, perfect fairness of DI = 1.0 can be achieved at a cost of the worst possible accuracy of ACC = 0.5. Depending on the method, the jump in disparate impact and accuracy score is rather abrupt or more fine-grained when reaching this extreme of $(DI, ACC) = (1.0, 0.5)$: Especially the COV- and the DI-methods relying on the optimization of fairness while constraining the accuracy using the hyperparameter $\lambda$ allow the latter, because the accuracy constraint is less sensitive than the covariance constraints, controlled by the hyperparameter $c$.[17]

However, also some of the TFPR- and the ACC-methods relying on the optimization of performance while constraining the fairness using the hyperparameter $c$ allow fine-grained variations in both scores. This motivates us to investigate the different methods also within the chosen subcategories. We do so in Appendix B.

Here, we find that the DI+ACC-ndb-max-method provides the best results while also providing the benefit of only requiring a few hyperparameters which are easy to choose. This finding makes the DI+ACC-ndb-max-method the best candidate to be tested on a more complicated and by that, more realistic, WDS, as we will do in subsection "L-town". However, before we do so, we investigate more the relation between the performance and fairness scores and the fairness-hyperparameters $c$ and $\lambda$.

*The influence of the fairness-hyperparameters on fairness and overall performance:* In Fig. 8, for the best-performing method of the TFPR-methods and the DI-methods—the results for the ACC-methods look similar to the ones of the TFPR-methods and the results for the COV-methods look similar to the ones of the DI-methods –, we show how the hyperparameters are related to disparate impact and accuracy, but this time, also equal opportunity score. Each of the three scores is plotted against the used hyperparameter of the corresponding fairness-enhancing method tested.

For the TFPR+COV-ndb-log-method (and the ACC+COV-ndb-log-method), the decrease of the hyperparameter $c$ is accompanied by the improvement of the fairness measures as well as the decrease of the performance measure. This can be explained as follows: A high empirical covariance between a sensitive feature and the prediction of the ensemble classifier means that the relative number of positive predictions within the related group differs significantly from the relative number of positive predictions within a group with small covariance. Thus, the more the covariance is constrained by the hyperparameter $c$, the less such extreme differences in the relative number of positive predictions across groups occur, leading to a better fairness score. In the case of disparate impact, therefore, a (better) higher score at the expense of a (worse) lower overall performance–compared to the overall performance that occurs in the unconstrained case or for a looser constraint, that is a larger bound by $c$,–appears. In the case of equal

[17] This is because too small choices of $c$ cause possible solutions with penalty or barrier function(s) of infinity. In such case, the trivial solution of only predicting leakages remains left, as in this case, the covariance becomes zero (Eq. (4.10) for $f^c(r(t_i), \Theta) = 1$ for all $i = 1, \ldots, n$) and by that, the penalty or the barrier function(s) are *not* infinity. Even more, in this case, all TPRs per group and by that, the disparate impact score, are equal to DI = 1. Moreover, in this case, the accuracy score is approximately ACC = 0.5 as the data set used is balanced with respect to the labels (Application Domain and Data Set). Therefore, in such cases, we end up with the trivial combination of $(DI, ACC) = (1.0, 0.5)$. In contrast, starting with a hyperparameter $\lambda$ that causes the trivial solution and then decreasing this hyperparameter to enforce a larger accuracy until the optimal accuracy is achieved ($\lambda = 0$) easily allows optimizing fairness without harming the accuracy constraint as it is measured in units of the optimal accuracy, that - as proven by the corresponding baseline method—exists.

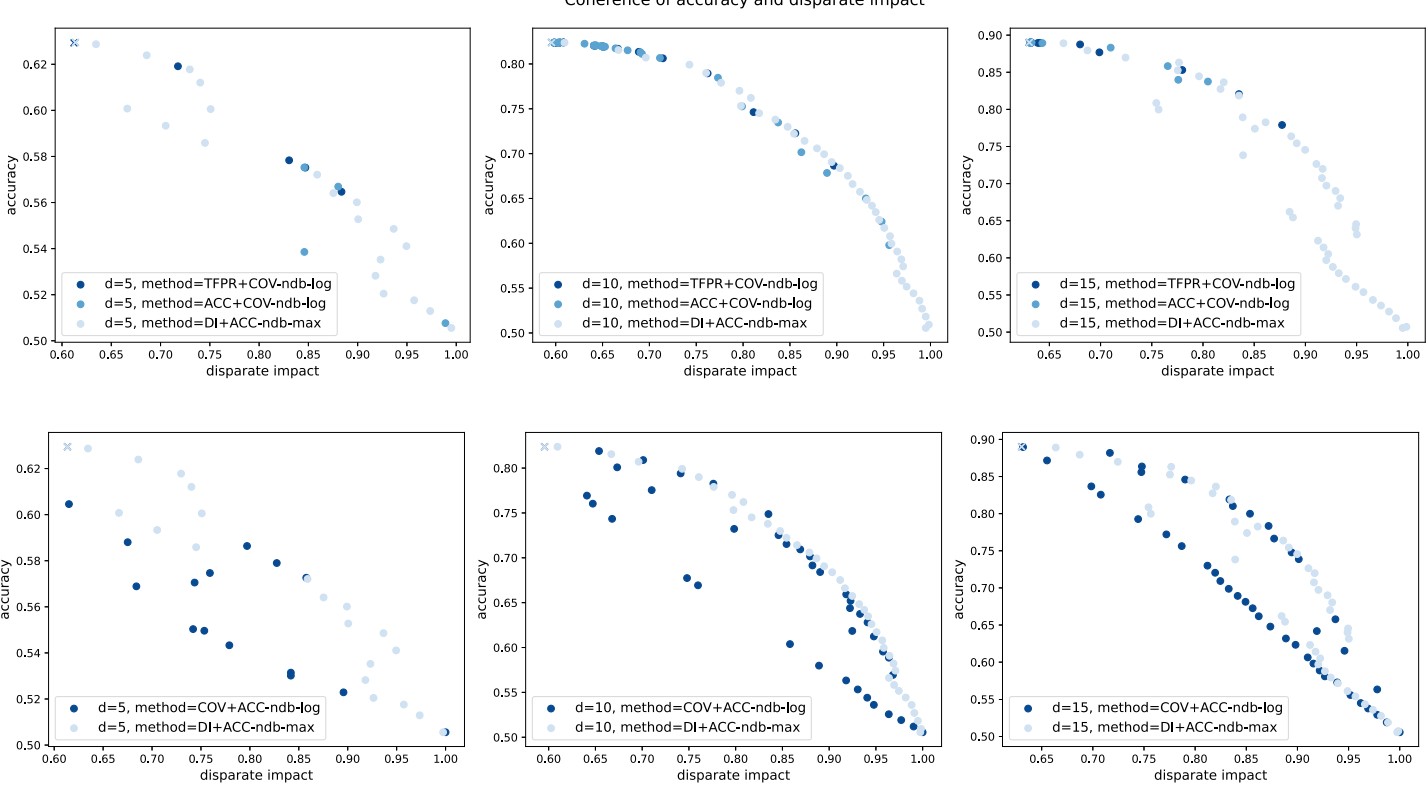

**Figure 7 Coherence of accuracy and disparate impact score for the different fairness-enhancing methods and different leakage sizes in the Hanoi-WDS, based on different hyperparameters $c$ or $\lambda$.** The cross data points visualize the accuracy and disparate impact score of the corresponding baselines methods (cf. paragraph "Explicit methods" in subsection "Fair leakage detection instantiations" or Table 4).

opportunity, however, a (better) lower score at the expense of a (worse) lower overall performance appears.

In contrast, for the DI+ACC-ndb-max-method (and the COV+ACC-ndb-log-method), the increase of the hyperparameter $\lambda$ is accompanied by the improvement of the fairness measures as well as the decrease of the performance measure due to the fact that a higher hyperparameter $\lambda$ allows a larger deviation of the optimal accuracy score. Thus, the feasible search space is extended and a worse accuracy is penalized less or not at all, so that the fairness score in the objective can be optimized to a larger extent.

*Equivalence of disparate impact and equal opportunity:* Moreover, especially to mention is the observation of our theoretical results from lemma 3.1 and corollary 3.2 in practice: For the coherence of equal opportunity score and the hyperparameters, the results in Fig. 8 equal the ones for disparate impact score in the same figure, but reflected along the horizontal axis through the point (0, 0.5). This proves the equivalence of both fairness measures as theoretically proven in lemma 3.1 and corollary 3.2. Nevertheless, note that this is an application specific result and does not hold in general.

Finally, as another new contribution compared to our previous work *Strotherm & Hammer (2023)*, we will test the best-performing DI+ACC-ndb-max-method on a more

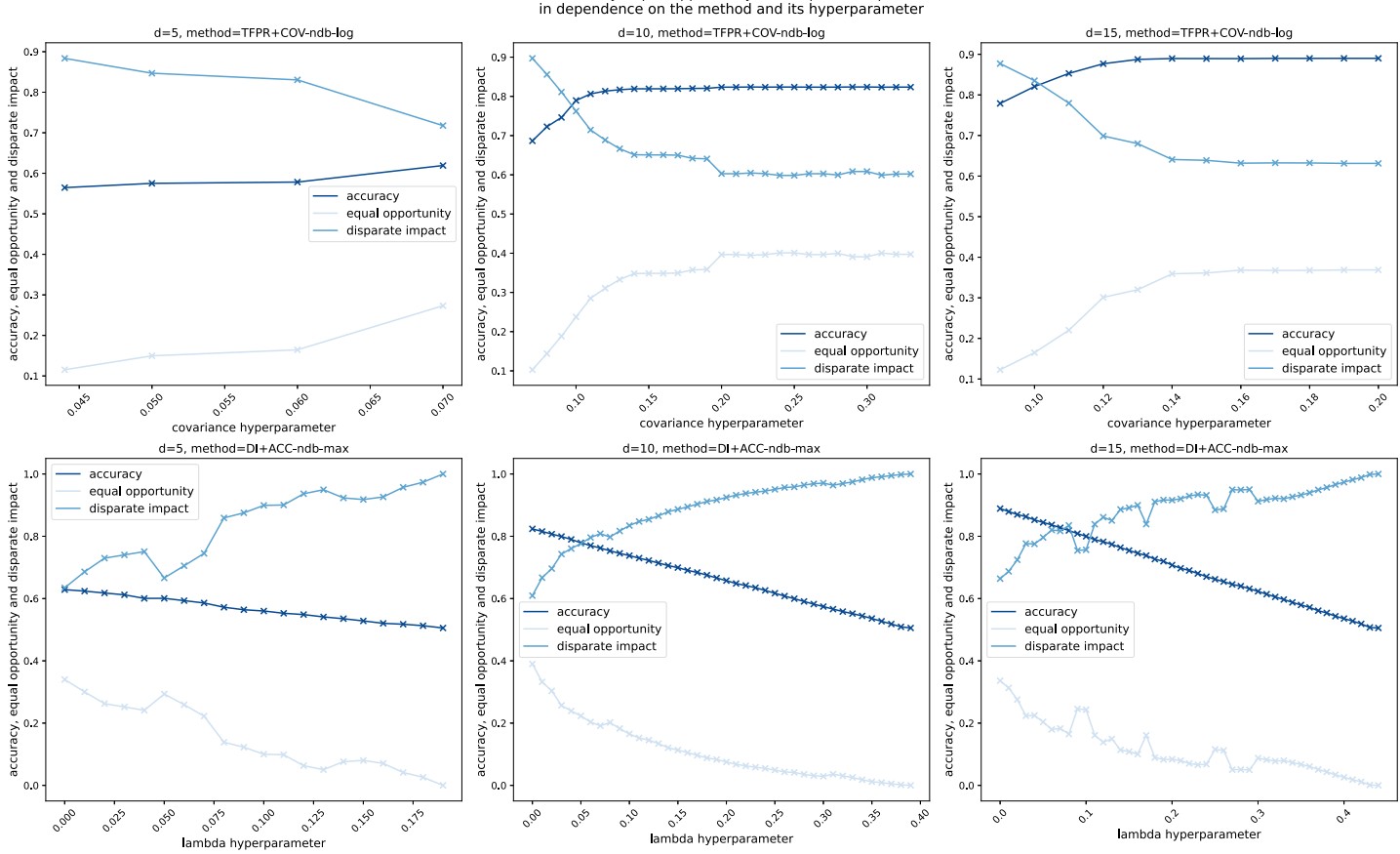

**Figure 8 Coherence of accuracy, disparate impact, equal opportunity and the training hyperparameter for different fairness-enhancing methods and different leakage sizes in the Hanoi-WDS.**

complex and by that realistic WDS, L-Town, using the more powerful GCN-virtual sensors incorporated into the leakage detection method.

### L-Town

**Initial parameters** While the dimension of the search space $\mathcal{X} = \mathbb{R}_+^d$ is equal to $d = 3$ (with $d$ the number of sensors) in Hanoi, it extends to $d = 33$ in L-Town (cf. subsection "Application domain and data set"). By that, chances are high that the graph of the objective function that needs to be optimized in each of the presented optimization-based methods (cf. subsection "Methodology of fairness-enhancing leakage detection") gets more complex and exhibits more saddle points and local minima. This intuition turns out to be true in practice, where the choice of the initial start parameters $\Theta_0 \in \mathcal{X}$ are crucial to the success of the methods tested. Therefore, for the experiments on the L-Town WDS, we use the hyperparameter $\Theta_{\text{opt.}} \in \mathcal{X}$ found for the H-method (cf. subsection "Methodology of leakage detection") only as an initial parameter $\Theta_0 \in \mathcal{X}$ for the ACC-ndb-method, which is the corresponding baseline method for the DI+ACC-ndb-max-method (paragraph "Explicit methods" in subsection "Fair leakage detection instantiations" or Table 4) that

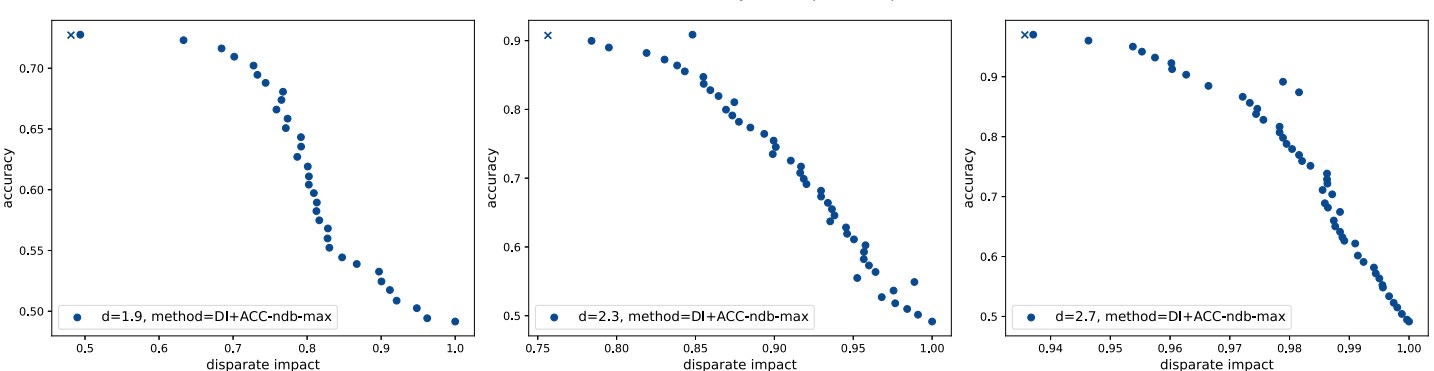

**Figure 9 Coherence of accuracy and disparate impact score for the DI+ACC-ndb-max-method and different leakage sizes in the L-Town-WDS.** The cross data points visualize the accuracy and disparate impact score of the corresponding baseline, the ACC-method.

turned out to work best in the previous subsection "Hanoi". Using the same initial parameter for the DI+ACC-ndb-max-method itself did not provide optimal results (–the pareto-front obtained here did not end up in the score combination of the corresponding baseline method). Therefore, consecutively, we use the hyperparameter $\Theta_{\text{opt.}} \in \mathcal{X}$ found by the ACC-ndb-method as an initial parameter $\Theta_0 \in \mathcal{X}$ for the DI+ACC-ndb-max-method.

**Hyperparameters** In view of Table 4, the ACC-ndb-method does not require the choice of any hyperparameters. For the DI+ACC-ndb-max-method, we vary the fairness-hyperparameter $\lambda \in [0, 1]$ and also choose $\mu = 100$ as discussed in subsection "Hanoi".

**Results** Similar to Fig. 7 for Hanoi, Fig. 9 shows the relation between the fairness and the overall performance of the trained model applied to L-Town.

The observations for L-Town are similarly well compared to those on Hanoi. Although while at first, it seems that there are less score combinations apart from, or more precisely, *below*, the pareto-front compared to the results of the same method applied to Hanoi, some score combinations *above* the seemingly optimal pareto-front may give rise to the existence of an even better pareto-front, which is not observed completely due to non-convexity of the OP.

Nevertheless, a desired disparate impact score of about 0.8 can be achieved by a decrease of accuracy by approximately 0.1 points for d = 1.9 and 0.01 points for d = 2.3 below the optimal accuracy obtained. For d = 2.7, the leakages are already almost detected perfectly and fair by the corresponding baseline ACC-ndb-method. Anyways, the fairness-enhancing DI-ACC-ndb-max-method is better by approximately 0.015 points in disparate impact score with barely no loss in accuracy.

Finally, similar to Fig. 8 for Hanoi, Fig. 10 shows how the hyperparameters are related to accuracy, disparate impact and equal opportunity score in the setting of L-Town. The results go hand in hand with the observations found for Hanoi, and also the equivalence between the two fairness scores can be observed again.
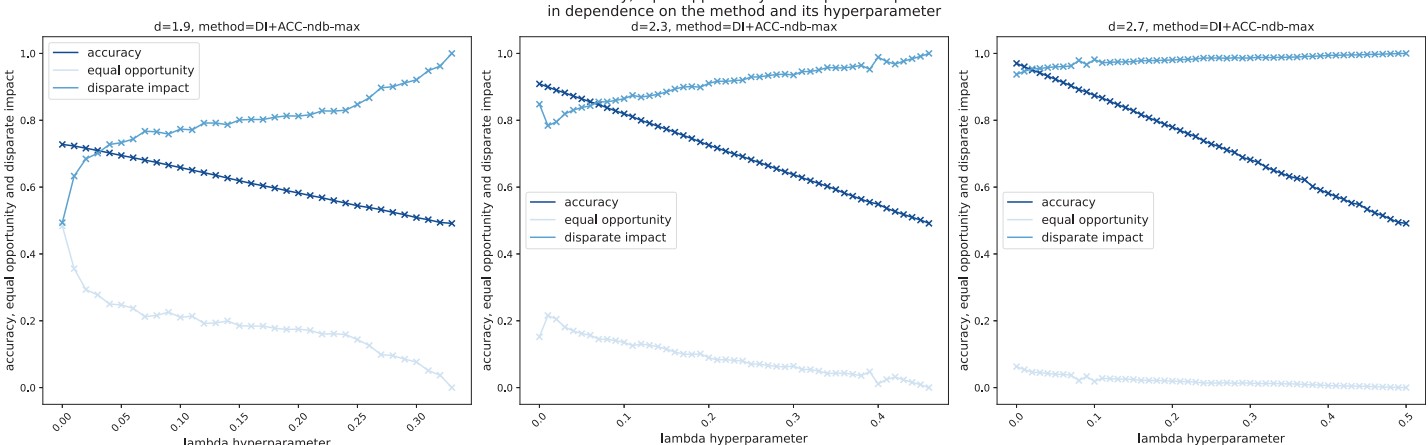

**Figure 10** Coherence of accuracy, disparate impact, equal opportunity and the training hyperparameter for the DI+ACC-ndb-max-method and different leakage sizes in the L-Town-WDS.

Additionally, we see by the position of the accuracy curves and the slope of the fairness curves that on the one hand, the better the model performs in general, measured by the accuracy score, the fairer the model is initially, and on the other hand, the harder it is to make the model even fairer.

## CONCLUSION

In this work, we introduced the notion of group fairness in an application domain of high social and ethical relevance, namely in the field of water distribution systems (WDSs). This required the generalization of common group fairness definitions to a single or possibly multiple non-binary sensitive feature(s). To do so, we gave a detailed introduction on the concept of group fairness based on the mathematical concept of independence, derived these generalized definitions from this concept and proved that they coincide with common group fairness definitions in the case of a binary sensitive feature and a binary classification task. We then investigated on the fairness issue in the area of leakage detection within WDSs. We showed that standard approaches are not fair in the context of different groups related to the locality within the network. As a remedy, we presented multiple methods that increase fairness of the leakage detection model with respect to the introduced fairness notion while satisfying the legal notions of disparate treatment and disparate impact simultaneously. We tested these methods not only on the Hanoi WDS, but also on the more complex and by that more realistic L-Town WDS. We empirically demonstrated that fairness and overall performance of the model are interdependent and the use of hyperparameters provides the ability to trade off fairness and overall performance. However, this trade-off lies in the responsibility of the policy maker.

From a practical perspective, this trade-off can be achieved by testing different hyperparameters during training, which requires multiple runs of training. Hereby, one limitation of the proposed methods is their non-convexity and scalability to larger

networks, which affects the training time. Future work could investigate this issue. Moreover, the fact that increasing the fairness of a model comes with a loss in accuracy leads to the question of whether this loss can be granted. While in leakage detection, in practice, detecting as many leakages as possible without observing false positives is a priority, there are further applications in the domain of WDS even more relevant for fairness. So far, tackling these use-cases has failed due to the lack of necessary data, which remains for future work. To conclude, the notion of fairness within the water domain is still at its beginning and further work on other cases of application within this domain is crucial.

## LIST OF ABBREVIATIONS

| | |
|---|---|
| **ACC** | Accuracy |
| **AI** | artificial intelligence |
| **DI** | disparate impact |
| **DP** | demographic parity |
| **EO** | equal opportunity |
| **EOs** | equalized odds |
| **EU** | European Union |
| **FPR** | false positive rate |
| **GCN** | graph convolutional network |
| **iff** | if and only if |
| **ML** | machine learning |
| **OP** | optimization problem |
| **TNR** | true negative rate |
| **TPR** | true positive rate |
| **WDS** | water distribution system |

### Funding

This work was supported by the European Research Council (ERC) under the ERC Synergy Grant Water-Futures (Grant agreement No. 951424). There was no additional external funding received for this study. The funders had no role in study design, data collection and analysis, decision to publish, or preparation of the manuscript.

### Grant Disclosures

The following grant information was disclosed by the authors:
European Research Council (ERC) under the ERC Synergy Grant Water-Futures: 951424.

### Competing Interests

The authors declare that they have no competing interests.

## Author Contributions

- Janine Strotherm conceived and designed the experiments, performed the experiments, analyzed the data, performed the computation work, prepared figures and/or tables, authored or reviewed drafts of the article, and approved the final draft.
- Inaam Ashraf performed the experiments, prepared figures and/or tables, and approved the final draft.
- Barbara Hammer conceived and designed the experiments, authored or reviewed drafts of the article, and approved the final draft.

## Data Availability

The code is available at GitHub and Zenodo:

- https://github.com/jstrotherm/FairnessInWDSs_extended/releases/tag/v1.0.0

- Janine Strotherm. (2024). jstrotherm/FairnessInWDSs_extended: Fairness in Water Distribution Networks (v1.0.0). Zenodo. https://doi.org/10.5281/zenodo.12699497.

## Supplemental Information

Supplemental information for this article can be found online at http://dx.doi.org/10.7717/peerj-cs.2317#supplemental-information.

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
