# Peer review of "Fairness-enhancing classification methods for non-binary sensitive features—How to fairly detect leakages in water distribution systems"

_PeerJ Computer Science, doi:10.7717/peerj-cs.2317_

## Round 0.1 · original submission · Major Revisions

The authors are invited to address all reviewers' comments and discuss, in the paper and their reply letter, the novelty with respect to previous papers published by them.

· Appeal

Appeal

last year and as part of the PeerJ Pub Award, we have been invited to submit an extended version of our paper "Fairness-Enhancing Ensemble Classification in Water Distribution Networks" published in the proceedings of the IWANN 2023 conference. We have been very surprised that this extended version (submission "Fairness-enhancing classification methods for non-binary sensitive features - How to fairly detect leakages in water distribution systems" to PeerJ Computer Science) was now rejected with the reason that the submission has similarities with our previous work. We give detailed information in the introduction of the submission on how the submitted extension differs from our previous work and also indicated this fact in the submission details under Declarations.

Can you please clarify what has gone wrong in the procedure?


· · Academic Editor

Reject

The submission has similarities with their previous work [reference 35], but they did not mention how their work differs from the referenced one. In this case, the authors must provide such clarification. Furthermore, the authors failed to compare more significant aspects with the referenced work. If their work is an extended or improved version of the referenced one, the authors should offer a clear explanation and comparison.

Reviewer 1 ·

Basic reporting

1- English grammar can be further improved.
2- More literature review should be added for comparative analysis.
3- Graphical Comparison should be added in Results.

Experimental design

1- The research is in accordance with the Aims and Scope of Journal.
2- Contributions of the work should be added as separate section after introduction.
3- Proposed Methodology should be elaborated graphically for further understanding.

Validity of the findings

1- The results and Discussion section could benefit more with detailed discussion.
2- Some bar graphs should be added for comparative analysis of results.

Reviewer 2 ·

Basic reporting

The manuscript, titled "Fairness-enhancing classification methods for non-binary sensitive features - How to fairly detect leakages in water distribution systems," is well developed; however, I have some concerns after a complete review of the paper, which are:
1. The abstract needs to be rewritten in significant writing, and the conclusion needs to include the percent improvement in the proposed work.
2. The significance or novelty of the study must be included in the introduction section of the proposed work.
3. The problem of limitations in the proposed work is missing; some points are included, but it would be beneficial to explicitly state the specific limitations faced, providing more context for readers unfamiliar with the background study.
4. The literature is missing the latest related work; authors need to include the state-of-the-art in the proposed work, e.g.

Strotherm, J., & Hammer, B. (2023, June). Fairness-Enhancing Ensemble Classification in Water Distribution Networks. In International Work-Conference on Artificial Neural Networks (pp. 119-133). Cham: Springer Nature Switzerland.

5. Figure 1 is very simple and having no titles?
6. The graphs are very simple; it would be better to compare your work to the state-of-the-art.
7. Minor typos and grammatical mistakes need correction before revision.

Experimental design

The graphs are very simple; it would be better to compare your work to the state-of-the-art.

Validity of the findings

The graphs are very simple; it would be better to compare your work to the state-of-the-art.

Reviewer 3 ·

Basic reporting

No comment

Experimental design

Very good

Validity of the findings

Experiments good

Additional comments

Good all

·

Basic reporting

I have the following concerns on this manuscript:
1- I am confused about the authors' intentions in this manuscript. Are they referring to fairness in terms of the number of detectors in WDS, fairness in the detection process itself (which seems unattainable), or an ethical principle of fairness in the prediction models?

2- Considering the authors' extensive experience in this field and the fact that this study builds upon their prior research, it would be beneficial for them to explore a different case study instead of focusing solely on the WDS, especially if they aim to achieve a broader generalization.

3- If the authors are primarily concerned with the ethical principle of fairness in predictive modeling, I suggest that they use "the detection of hate speech in large language models" as a case study to demonstrate whether these models exhibit fairness or not.

Experimental design

1- I would like to inquire about how the sensors are distributed in the WDS. Are they part of a wireless sensor network?

2- Do all neural networks in your model use linear activation functions? If so, please clarify the reasoning behind this choice.

Validity of the findings

It is recommended to compare your findings with those of other researchers (state-of-the-art).

Additional comments

Ensure proper adherence to the acronym protocol in your manuscript. Is there a distinction between GCNs and GNNs?
Use "machine learning (ML)" the first time you mention ML.

---

## Round 0.2 · accepted · Accept

Congratulations, your paper is ready for publication

Reviewer 2 ·

Basic reporting

Author sufficiently enhanced their paper, so i accept it for publication.

Experimental design

N/A

Validity of the findings

N/A

·

Basic reporting

My previous concerns are partially addressed.

Experimental design

My previous concerns are partially addressed.

Validity of the findings

My previous concerns are partially addressed.

Additional comments

My previous concerns are partially addressed.